**RESEARCH**

# Histone modifications during the life cycle of the brown alga *Ectocarpus*

Simon Bourdareau[1], Leila Tirichine[2], Bérangère Lombard[3], Damarys Loew[3], Delphine Scornet[1], Yue Wu[2], Susana M. Coelho[1,4*] and J. Mark Cock[1*]

\* Correspondence: susana.coelho@
tuebingen.mpg.de; cock@sb-roscoff.
fr
[1]CNRS, Sorbonne Université, UPMC
University Paris 06, Algal Genetics
Group, UMR 8227, Integrative
Biology of Marine Models, Station
Biologique de Roscoff, CS 90074,
F-29688 Roscoff, France
Full list of author information is
available at the end of the article

## Abstract

**Background:** Brown algae evolved complex multicellularity independently of the animal and land plant lineages and are the third most developmentally complex phylogenetic group on the planet. An understanding of developmental processes in this group is expected to provide important insights into the evolutionary events necessary for the emergence of complex multicellularity. Here, we focus on mechanisms of epigenetic regulation involving post-translational modifications of histone proteins.

**Results:** A total of 47 histone post-translational modifications are identified, including a novel mark H2AZR38me1, but *Ectocarpus* lacks both H3K27me3 and the major polycomb complexes. ChIP-seq identifies modifications associated with transcription start sites and gene bodies of active genes and with transposons. H3K79me2 exhibits an unusual pattern, often marking large genomic regions spanning several genes. Transcription start sites of closely spaced, divergently transcribed gene pairs share a common nucleosome-depleted region and exhibit shared histone modification peaks. Overall, patterns of histone modifications are stable through the life cycle. Analysis of histone modifications at generation-biased genes identifies a correlation between the presence of specific chromatin marks and the level of gene expression.

**Conclusions:** The overview of histone post-translational modifications in the brown alga presented here will provide a foundation for future studies aimed at understanding the role of chromatin modifications in the regulation of brown algal genomes.

**Keywords:** Brown algae, ChIP-seq, Chromatin, *Ectocarpus*, Gametophyte, Histone modification, Life cycle, Multicellularity, Polycomb complex, Sporophyte

## Introduction

Very few eukaryotic lineages have independently evolved complex multicellularity, and the brown algae are of particular interest as the third most developmentally complex lineage after animals and land plants. The size and complexity attained by some brown

 

algae are illustrated by the giant kelp *Macrocystis*, which can reach up to 50 m in length.

The deployment of developmental programmes and the establishment of different cell fates in complex multicellular organisms imply the acquisition of different epigenetic states in different cell types. A number of mechanisms underlie the establishment and maintenance of epigenetic states. These include, for example, cell signalling and regulation of gene transcription, but another important factor is the chemical modification of chromatin. In particular, histones have been shown to undergo a broad range of different post-translational modifications (PTMs) involving the addition of various chemical moieties to multiple amino acid residues, particularly within the unstructured amino-terminal "tails" of these proteins [1]. These modifications affect chromatin function either by directly modifying interactions between the different components of the nucleosome or via the action of proteins that bind to specific histone modifications and effect specific tasks. In this manner, histone PTMs act as a "histone code", mediating the establishment and maintenance of different epigenetic states across the genome.

Epigenetic processes not only allow cells that carry the same genomic information to assume different functions in different parts of a multicellular organism but they also allow temporal changes, i.e. establishment of different cellular functions at different stages of the life history. This latter aspect is particularly important in organisms that have complex life cycles. For example, many brown algae have haploid-diploid life cycles involving an alternation between two different organisms, the sporophyte and the gametophyte, often with very distinct body plans [2]. These life cycles imply complex regulation of chromatin, but at present, very little is known about the role of histone PTMs, for example, in the regulation of developmental and life cycle processes in the brown algae.

The filamentous alga *Ectocarpus* is being used as a model system to study brown algal developmental biology [3–7]. *Ectocarpus* has a haploid-diploid life cycle involving an alternation between a gametophyte, which is usually haploid, and a sporophyte, which is usually diploid. However, there is clear evidence that the identity of each life cycle generation is not determined by its ploidy because haploid sporophytes (partheno-sporophytes) can be produced by parthenogenetic development of haploid gametes [8, 9], and diploid gametophytes can be constructed using mutants that are unable to deploy the sporophyte developmental pathway [3, 7]. These observations indicate that epigenetic processes play an important role during the *Ectocarpus* life cycle. Recent work has shown that the deployment of the sporophyte programme requires two different three-amino acid loop extension homeodomain transcription factors (TALE HD TFs), OUROBOROS (ORO) and SAMSARA (SAM) [7]. Remarkably, TALE HD TFs appear to have been recruited convergently to regulate sporophyte development in both the brown algal and the land plant lineages [7]. In land plants, the PRC2 polycomb complex has also been implicated in life cycle regulation [10–17], indicating that chromatin modification processes play an important role in life cycle regulation in that lineage. Chromatin modification has been proposed to play a similar role in the brown algae [2], but this hypothesis has not been investigated experimentally.

In this study, we have carried out a broad census of histone PTMs in *Ectocarpus* chromatin and have developed a method to evaluate the genome-wide distribution of specific histone PTMs in this species. Application of this method allowed the

identification of histone PTMs associated with transcriptional start sites (H3K4me2, H3K4me3, H3K9ac, H3K14ac and H3K27ac) and gene bodies (H3K36me3) of actively transcribed genes and a histone PTM associated with transposons and repetitive sequences (H4K20me3). We also show that H3K79me2 often marks extensive regions spanning several genes and suggest some possible functions for this PTM. Overall, genome-wide histone PTM patterns were found to remain stable following transition between the sporophyte and gametophyte generations of the life cycle, consistent with the observation that only 4% of genes exhibited a significant level of generation-biased expression. Analysis of generation-biased genes allowed changes in chromatin state (combinations of histone PTMs) to be correlated with changes in gene expression.

## Results

### *Ectocarpus* histones and histone-modifier enzymes

Analysis of the *Ectocarpus* genome sequence [18] identified 34 core histone and nine H1 histone genes (Additional file 5: Table S1). Four of the core histone genes are predicted to encode variant forms, including probable CenH3, H2A.X and H2A.Z proteins. All nine H1 histone genes appear to encode bona fide H1 proteins and are not members of related families such as the plant GH1-HMGA or GH1-Myb families [19]. All but eight of the histone genes were located in five gene clusters on chromosomes 4, 7 and 26 and on an unmapped scaffold (sctg_442). The organisation of the clusters suggests multiple duplication, rearrangement and fragmentation of an ancestral cluster with the organisation H4, H1, H3, H2B and H2A (Additional file 2: Figure S1).

A search for genes encoding histone-modifying enzymes identified both putative histone acetyltransferases and methyltransferases, and predicted deacetylase and demethylase enzymes (Additional file 6: Table S2). Most of the acetyltransferases were sufficiently similar to well-characterised homologues to allow prediction of their target residues, but the methyltransferases tended to be less conserved at the sequence level and, in many cases, had novel domain structures. Direct functional information, for example based on mutant analysis, will therefore be necessary to investigate the specificity of the *Ectocarpus* methyltransferases.

### Identification of histone PTMs in *Ectocarpus*

Histone PTMs were detected using mass spectrometry of enzyme-digested histone preparations. In addition, a broad range of commercially available antibodies was tested against *Ectocarpus* histone preparations on immunoblots to further confirm the presence of a subset of the PTMs identified by mass spectrometry. A total of 47 PTMs of core and variant histones were detected in *Ectocarpus* (Fig. 1a, Additional file 3: Figure S2). Six of these marks were only detected by immunoblotting and should therefore be treated with caution (marked with an asterisk in Fig. 1a, Additional file 7: Table S3). Note that two of these PTMs, H3K9me2 and H3K9me3, were also not detectable using mass spectrometry in *Arabidopsis* but have been detected using immunoblotting [20, 21]. Figure S3 (Additional file 4) shows immunoblots that detected a weak signal for these two PTMs in *Ectocarpus* chromatin.

Most of the histone PTMs detected in *Ectocarpus* have been reported previously in species from one or more of the land plant, animal or fungal lineages, either at exactly

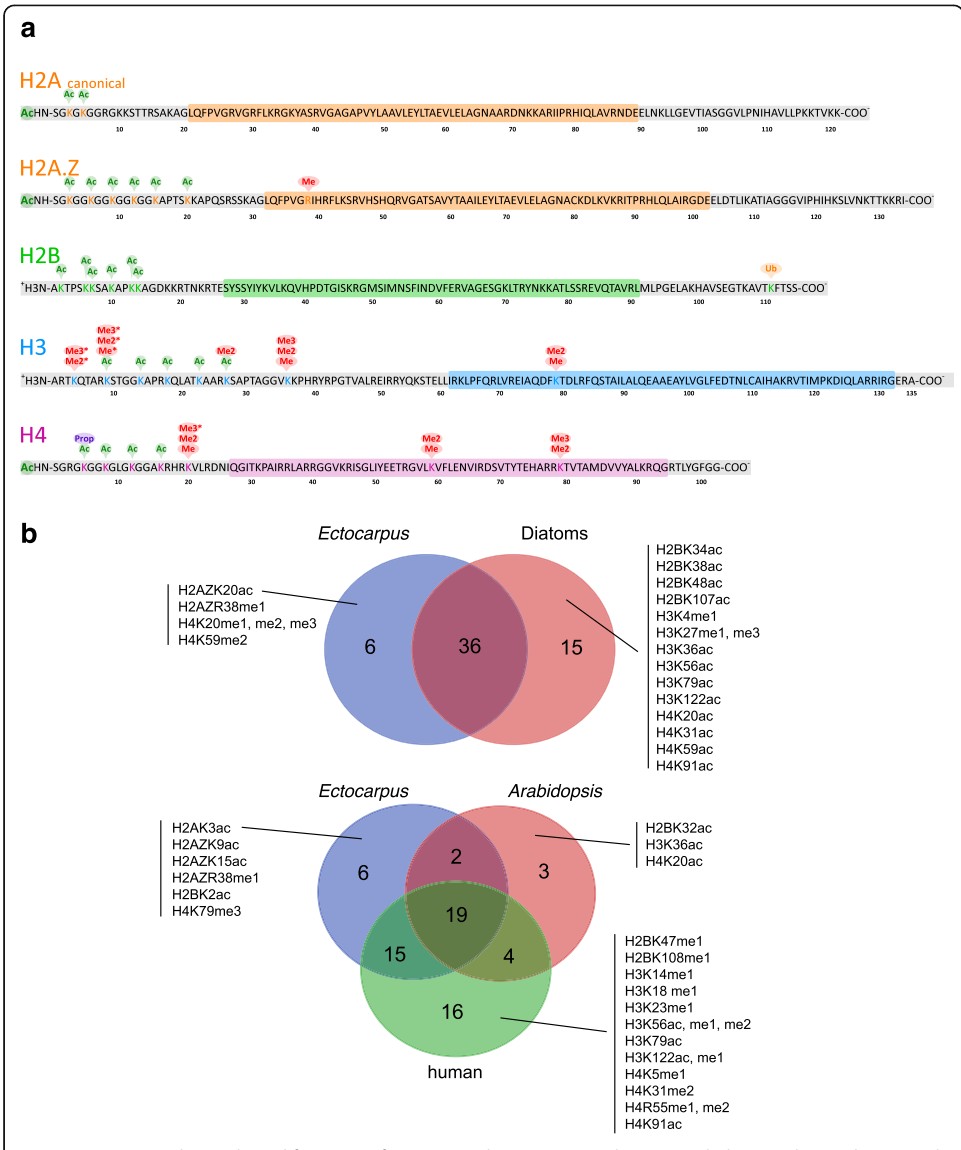

**Fig. 1 a** Post-translational modifications of *Ectocarpus* histones. Acetylation, methylation, ubiquitylation and propionylation modifications of core and variant histones identified in this study. Coloured boxes indicate globular core domains, and grey boxes indicate N- and C-terminal tails. Amino acid positions are indicated below the protein sequence. Asterisks indicate histone modifications that were only detected using antibodies. All other modifications were detected by mass spectrometry. **b** Comparison of the histone PTMs detected in *Ectocarpus* with PTMs reported for diatoms (combined data for *Phaeodactylum tricornutum* and *Thalassiosira pseudonana*, upper panel) and for humans and the land plant *Arabidopsis* (lower panel) (see Additional file 5: Table S3 for details)

the same position or at an equivalent position in the corresponding protein (Fig. 1b, Additional file 7: Table S3). However, a number of marks have only been described so far in stramenopiles. Of these, some PTMs such as acetylation of lysines 9 and 15 of H2A.Z were detected in both *Ectocarpus* and the diatom *Phaeodactylum tricornutum* [22], whereas others were detected only in the diatoms (e.g. acetylation of lysine 59 on H4) or only in *Ectocarpus* (methylation of arginine 38 on H2A.Z; Fig. 1b).

### Genome-wide distribution of selected histone PTMs

To investigate the functions of histone PTMs in *Ectocarpus*, we analysed the distribution of eight selected marks across the genome using chromatin immunoprecipitation and sequencing (ChIP-seq). The PTMs analysed were H3K4me2, H3K4me3, H3K9ac, H3K14ac, H3K27ac, H3K36me3, H3K79me2 and H4K20me3 (Fig. 2). Five of these PTMs, H3K4me2, H3K4me3, H3K9ac, H3K14ac and H3K27ac, were preferentially associated with the transcription start sites (TSSs) of genes but tended to be depleted from gene bodies (Fig. 2a–d). Genome-wide patterns of these five PTMs exhibited a range of levels of positive correlation (Pearson's *r* 0.13 to 0.86; Fig. 2e), indicating that they tended to be co-localised in the genome. Similar PTM patterns were detected at protein-coding and lncRNA genes (Fig. 2c), suggesting that histone PTMs may be used in a similar manner to regulate these two types of gene. For each of the five TSS-associated PTMs, a peak was detected within 500 bp of the TSS for between 77 and 83% of the genes in the genome. For H3K4me2, a pair of adjacent peaks was detected, one on each side of the TSS (Fig. 2b, c). Genome-wide, peaks of H3K4me2, H3K4me3, H3K9ac, H3K14ac and H3K27ac marks were associated with between 13,416 and 14,423 of the 17,406 genes. We noted a strong positive correlation between gene expression level, measured in transcripts per million (TPM), and the strength of mapping of each mark to TSSs (Fig. 3). Taken together, these observations indicate a strong association of H3K4me2, H3K4me3, H3K9ac, H3K14ac and H3K27ac with transcriptionally active genes in *Ectocarpus*.

The three remaining PTMs, H3K36me3, H3K79me2 and H4K20me3, were depleted from TSSs (Fig. 2). H3K36me3, which was detected at 12,863 genes (74% of the genome), was most strongly associated with gene bodies and was depleted from both TSSs and transcription end sites (TESs). As observed for the TSS-localised PTMs discussed above, the presence of H3K36me3 on gene bodies was positively correlated with expression, indicating that this mark is also associated with gene activation (Fig. 3).

H4K20me3 peaks were detected principally in the non-coding part of the genome (intergenic regions and introns; Fig. 2a) and occurred preferentially in intergenic regions or introns that also contained a transposon (Pearson's $\chi^2$ test with Yates' continuity correction *p* value $< 2.2e^{-16}$ for both introns and intergenic regions). Indeed, almost all (94.6%) of the H4K20me3 peaks co-localised with either a transposon or a region of a short repeated sequence. The distribution of H4K20me3 was therefore consistent with it playing a role in the regulation of inserted transposons, possibly in the silencing of these elements. H4K20me3 peaks were associated with all types of transposon, including both class I (RNA) and class II (DNA) transposons (Additional file 5: Table S4). The occurrence of H4K20me3 in introns was not unexpected because introns make up approximately 40% of the *Ectocarpus* genome [23] and therefore represent a significant proportion of the non-coding fraction of the genome. Moreover, 44.0% of the transposons (repeated elements ≥ 400 bp) in the genome are located in introns.

Genes marked with H4K20me3 tended to have fewer exons, to be longer and to have lower levels of expression than genes without a H4K20me3 peak (Fig. 4a–c). The inverse relationship between the presence of H4K20me3 and gene expression level (Fig. 3) could be interpreted as indicating a role in gene regulation, but given the co-localisation of this PTM with transposon sequences, a more likely explanation is that the observed effect on gene expression was an indirect effect of silencing of intronic

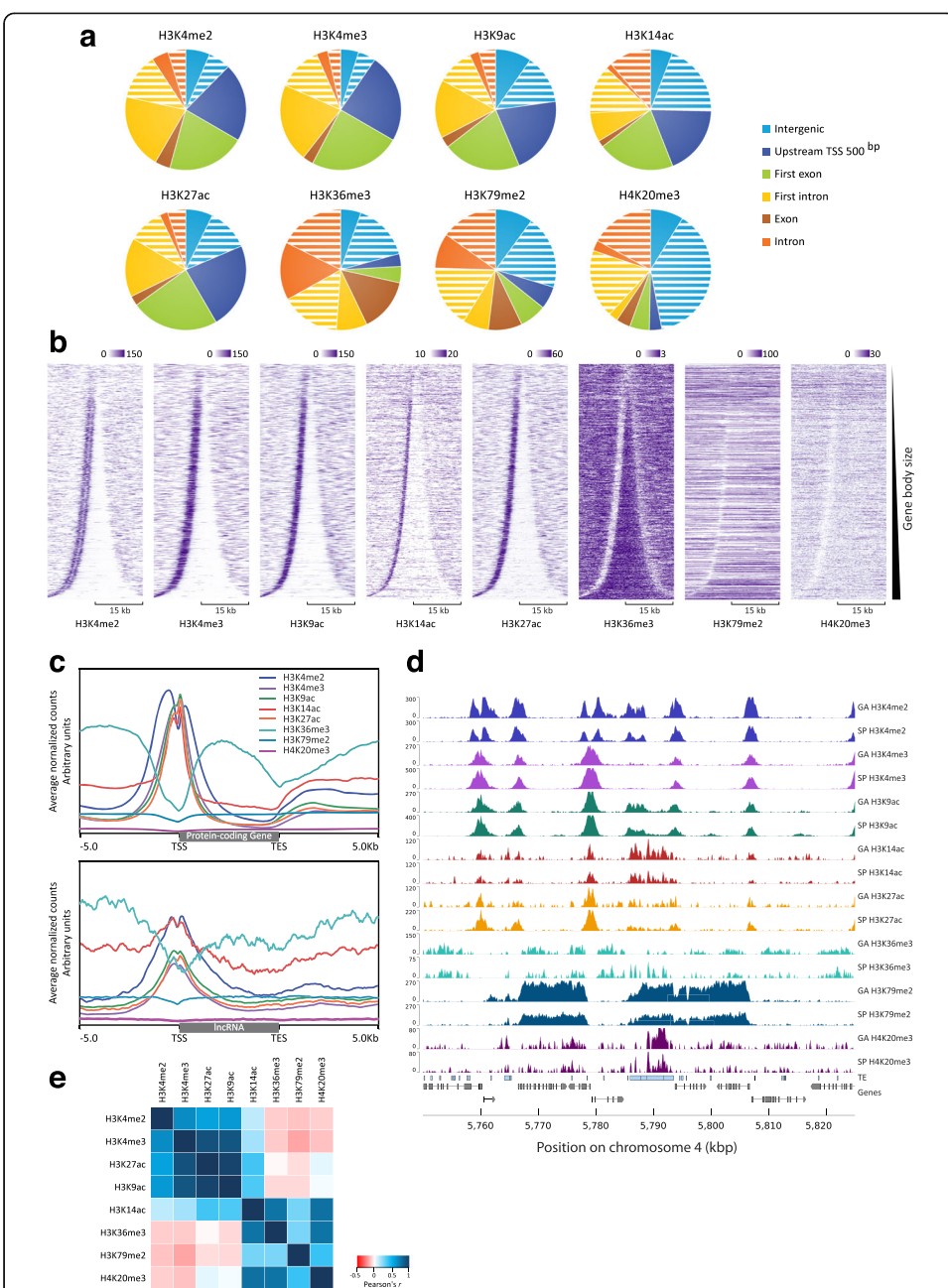

**Fig. 2** Distribution of histone PTMs in the *Ectocarpus* genome. **a** Distribution of eight histone PTMs across six genome feature classes. Intergenic, all intergenic regions except the TSS regions; upstream TSS 500 bp, 500 bp region 5′ to the transcription start site; exon, all exon sequence except first exons; intron, all intron sequence except first introns. The sectors for intergenic, intron and first intron regions have been separated into two to indicate values for regions that contain TEs (white stripes) and regions that do not (full colour). **b** Distribution of the eight histone PTMs across the complete set of *Ectocarpus* genes (17,447) sorted by gene body size and centred on the gene body. The black brackets correspond to 15 kb. **c** Enrichment profiles of the eight histone PTMs for all protein-coding (top panel) and all lncRNA (bottom panel) genes. Gene bodies are plotted as proportional lengths, upstream and downstream intergenic regions in kilobases. **d** Representative region of chromosome 4 showing profiles of mapped ChIP-seq reads for the eight histone PTMs in the sporophyte (SP) and gametophyte (GA) generations. Light blue boxes represent transposons (TEs). Grey boxes and arrows represent exons superimposed on genes represented by a black line. **e** Pearson correlation scores for comparisons of the genomic distributions of ChIP-seq signal peaks for the eight histone PTMs

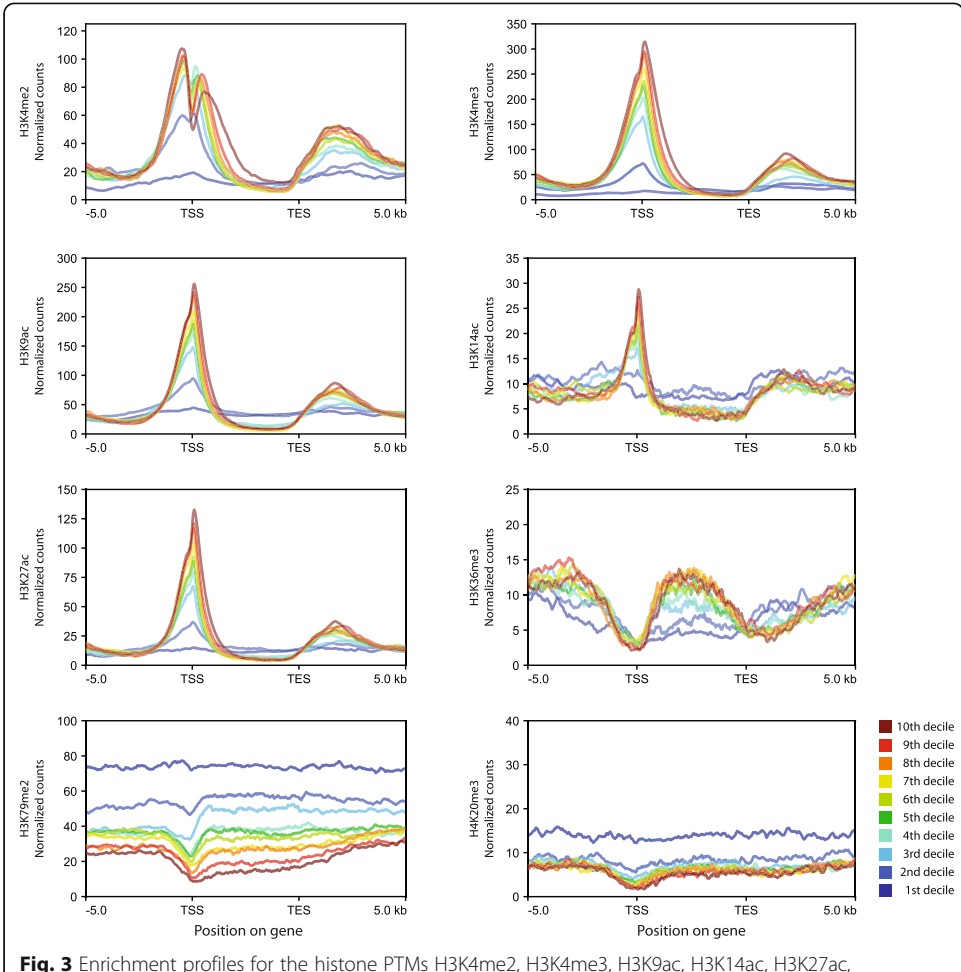

**Fig. 3** Enrichment profiles for the histone PTMs H3K4me2, H3K4me3, H3K9ac, H3K14ac, H3K27ac, H3K36me3, H3K79me2 and H4K20me3 across genes expressed at different levels (TPM deciles). Gene bodies are plotted as proportional lengths, upstream and downstream intergenic regions in kilobases. TSS, transcription start site; TES, transcription end site; kb, kilobases

transposon sequences. Interestingly, genes that were marked with H4K20me3 were significantly less strongly marked with the TSS-associated PTMs H3K4me2, H3K4me3, H3K9ac, H3K14ac and H3K27ac (Fig. 4d). Several hypotheses could be proposed to explain this observation. If the presence of H4K20me3 within a gene directly causes a reduction in the abundance of TSS-associated PTMs, this may constitute part of the mechanism that mediates the repressive effect of H4K20me3 on gene expression. Alternatively, it is also possible that the reduction in the abundance of TSS-associated PTMs is an indirect consequence of H4K20me3-induced repression of gene expression.

## The genome is partitioned into H3K79me2-marked and H3K79me2-depleted regions

H3K79me2 exhibited an unusual distribution pattern. This PTM was detected throughout the genome, often in large, discrete, continuously marked regions of several kilobases separated by H3K79me2-depleted regions (Fig. 5a, Additional file 4: Figure S4). Together, the H3K79me2 regions covered 74.5 Mbp, 37.2% of the genome. About a third (36.6%) of the H3K79me2 regions were longer than 5 kbp, and many of these

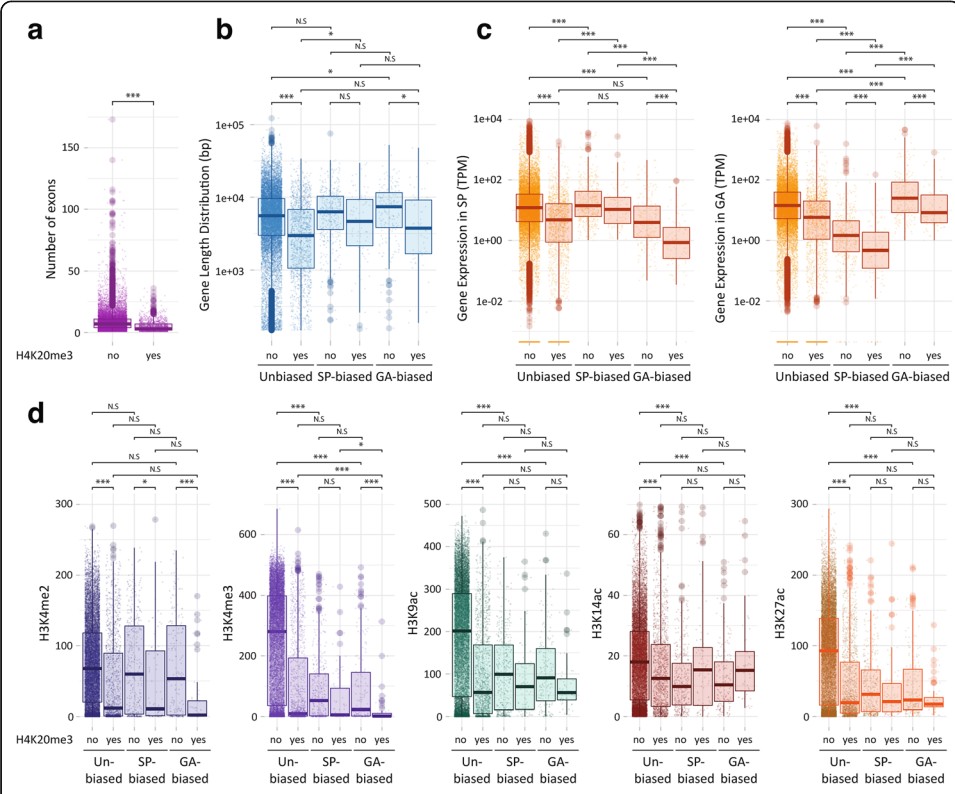

**Fig. 4** Comparison of genes that contained or did not contain a H4K20me3 peak. **a** Genes with at least one H4K20me3 peak (yes) have fewer exons than genes without a H4K20me3 peak (no). **b** Lengths of genes with or without a H4K20me3 peak. **c** Abundances of transcripts for genes with or without a H4K20me3 peak. **d** Histone PTMs on genes with or without a H4K20me3 peak. SP, sporophyte; GA, gametophyte. Asterisks indicate significant differences (Bonferonni-adjusted Wilcoxon tests; ***$p < 0.0001$; *$p < 0.05$)

regions spanned more than one gene (on average, H3K79me2 regions of > 5 kbp contained 1.85 genes). The borders of the long H3K79me2 regions tended to be localised near a TSS or a TES. When H3K79me2 regions of > 5 kbp were considered, 49.3% of the borders were located within a kilobase of a TSS or a TES, and a Bonferonni-adjusted Wilcoxon test indicated that the borders were significantly closer to the nearest TSS or TES than they were to random anchor points in the genome (Additional file 4: Figure S5).

Comparison of ChIP-seq data for the sporophyte and gametophyte generations of the life cycle indicated that the H3K79me2 regions were stably maintained throughout the life cycle. Only 0.3% or 1.2% of H3K79me2 regions longer than 5 kbp were detected uniquely in the sporophyte or the gametophyte generation, respectively.

The H3K79me2 regions were strongly associated with peaks of H4K20me3 (Fig. 5a, b; Additional file 4: Figure S4). Consistent with the co-localisation of the two PTMs, genes within the H3K79me2 regions were significantly shorter, possessed fewer introns and their transcripts were less abundant on average than genes that were outside the H3K79me2 regions (Fig. 4c–e). The TSSs of genes within the H3K79me2 regions were also significantly less strongly marked by H3K4me2, H3K4me3, H3K9ac, H3K14ac and H3K27ac (Fig. 4f), and there was a negative correlation between gene expression level (measured in TPM) and the quantitative presence of H3K79me2 over genes (Fig. 3). To determine whether the association of H3K79me2 with reduced levels of gene

expression and reduced deposition of TSS-localised PTMs was an indirect consequence of the co-localisation of this mark with H4K20me3, we analysed loci (Fig. 5b) where the two PTMs occurred independently of each other (Additional file 4: Figure S6). This analysis indicated that both H3K79me2 and H4K20me3 were associated with reduced levels of gene expression and reduced deposition of TSS-localised PTMs, even when one PTM was present independently of the other. Moreover, the effects of H3K79me2 and H4K20me3 on gene expression and on the deposition of TSS-localised PTMs were additive when both PTMs were present at the same locus, i.e. levels of gene expression and TSS-localised PTMs were even lower when both PTMs were present. Therefore, like H4K20me3, H3K79me2 appears to be associated with the repression of the gene expression. However, based on a similar argument to that proposed for H4K20me3 above, the organisation of deposition of H3K79me2 across blocks of contiguous genes (Fig. 5) does not appear to be consistent with a direct role in the gene regulation, and we therefore favour the hypothesis that, as proposed for H4K20me3, the effect of H3K79me2 on gene regulation is an indirect consequence of the involvement of this PTM in another, currently unknown, role in chromatin homeostasis.

Chromosome 6 contains an integrated copy of a large DNA virus (spanning 0.3 Mbp from approximately 4,244,200 to 4,547,200) that is closely related to the *Ectocarpus* phaeovirus EsV-1. The inserted virus has been shown to be transcriptionally silent [18, 23]. A large H3K79me2 region of about 0.42 Mbp was detected that spanned the entire inserted viral genome (Additional file 4: Figure S7). Chromatin within this H3K79me2 region was also marked with H4K20me3, but the other histone PTMs assayed were depleted from the inserted viral genome (Additional file 4: Figure S7).

### Overlapping TSS regions of divergently transcribed gene pairs

One unusual feature of the *Ectocarpus* genome, compared to genomes of similar size, is that there is a relatively strong tendency for adjacent genes to be transcribed from opposite strands of the DNA helix [23]. Consequently, the genome contains many divergently transcribed gene pairs. To investigate the effect of this pattern of gene organisation on the chromatin characteristics of TSSs, we searched for pairs of adjacent genes located on the same sequence scaffold. This search identified 10,399 TSS-adjacent intergenic regions, and 61.7% of the genes flanking these regions were part of a divergently transcribed gene pair. The intergenic regions of divergently transcribed gene pairs were significantly shorter than those of tandem gene pairs (median 409 and 2293, respectively, Wilcoxon test, $p$ value $< 2e^{-16}$; Fig. 6a). When the intergenic regions of divergent gene pairs were shorter than about 600 bp, the two TSS chromatin regions overlapped and shared the same nucleosome-depleted region (NDR) based on micrococcal nuclease digestion data (Fig. 6a). This overlap correlated with the presence of double peaks for the PTMs H3K4me3, H3K9ac, H3K14ac and H3K27ac, one on each side of the TSS (Fig. 6a, b). For H3K4me2, which was detected as a double peak at most individual TSSs (Fig. 2b), the two peaks were further apart at the overlapping TSSs of divergent gene pairs than they were at the single intergenic TSSs of tandem gene pairs (Fig. 6a, b). Therefore, the pattern of PTMs also indicated that the TSSs of proximate divergent genes are located within a shared chromatin domain. Signals for all five of the above PTMs were significantly stronger for divergent than for tandem gene pairs

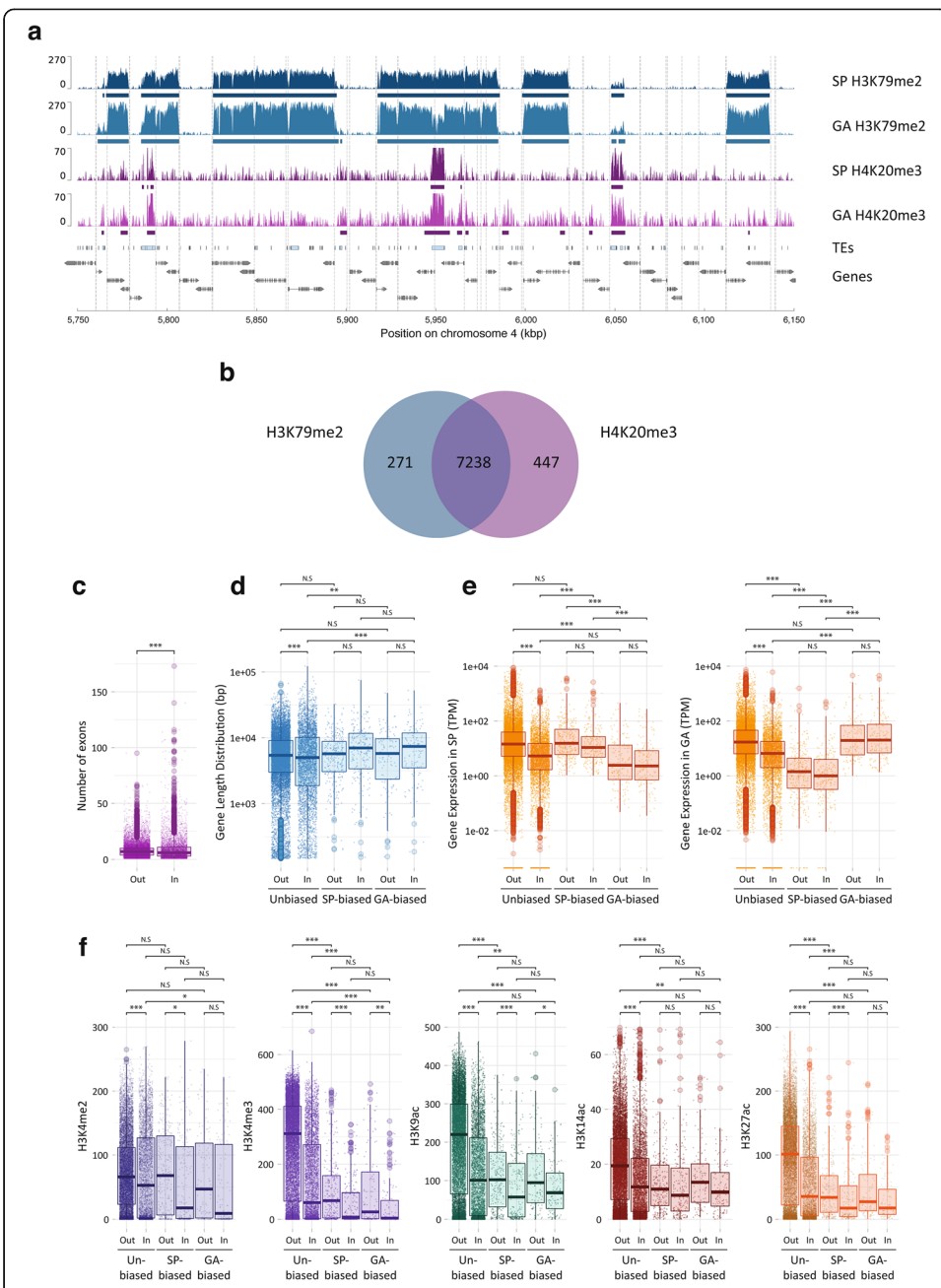

**Fig. 5** Genomic distribution of the histone PTM H3K79me2. **a** Representative region of 0.4 Mbp from chromosome 4 (spanning the genes with LocusIDs Ec-04_005800 to Ec-04_006210) showing alternating regions marked or not with H3K79me2 (blue peaks) during both the sporophyte and gametophyte generations of the life cycle. Blue bars indicate the H3K79me2 regions. The distribution of H4K20me3 (purple peaks) is also shown for comparison (peaks called by MACS2 are indicated by purple bars). Light blue boxes represent transposons (TEs). Grey boxes and arrows represent exons superimposed on genes represented by a black line. Vertical dotted lines indicate the positions of TSSs. **b** Genome-wide overlap between peaks of H3K79me2 and H4K20me3. **c** Genes in H3K79me2 regions have fewer exons. **d** Lengths of genes within (In) or outside (Out) H3K79me2 regions. **e** Abundances of transcripts for genes within (In) or outside (Out) H3K79me2 regions. **f** Histone PTMs on genes within (In) or outside (Out) H3K79me2 regions. A gene was considered to be "In" if more than 50% of the gene body (TSS-TES) was inside a H3K79me2 region. SP, sporophyte; GA, gametophyte; kb, kilobase pairs. Asterisks indicate significant differences (Bonferonni-adjusted Wilcoxon tests; ***$p < 0.0001$; **$p < 0.001$; *$p < 0.05$)

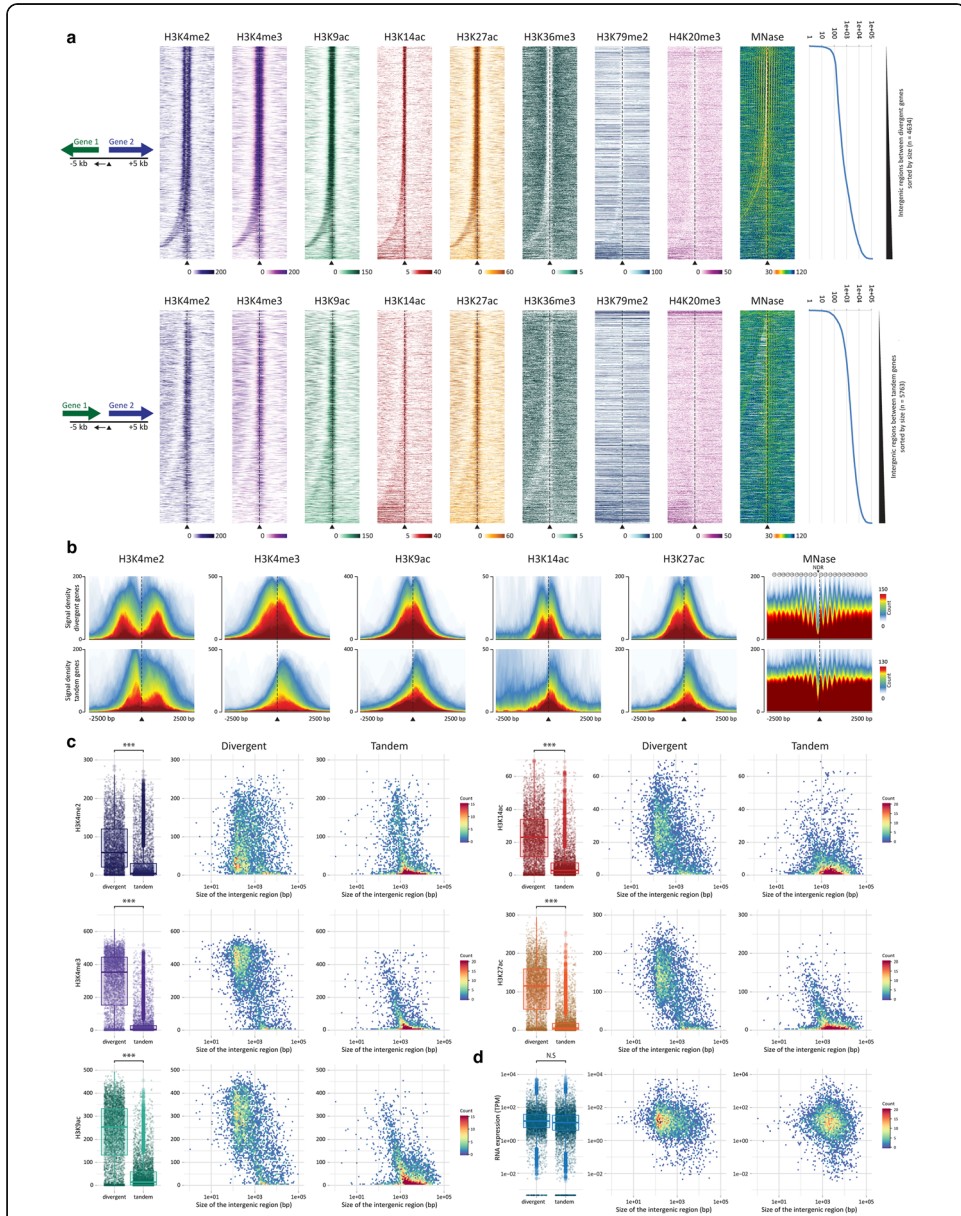

**Fig. 6** Histone PTM distributions in intergenic regions between divergent and tandemly organised pairs of genes. **a** Distribution of eight histone PTMs and MNase-sensitive sites in regions of 10 kb surrounding the TSS of the right-hand gene for pairs of either divergently (upper panel) or tandemly (lower panel) organised genes. The lengths of the intergenic regions are plotted on the right in base pairs. **b** Signal distributions for five TSS-localised histone PTMs and MNase-sensitive sites in regions of 5 kb surrounding the TSS of the right-hand gene for pairs of either divergently (upper plots) or tandemly (lower plots) organised genes on chromosome 4. **c** Abundance of five histone PTMs at the TSS of Gene 2 (see schema in **a**) in the intergenic regions of divergently or tandemly organised gene pairs together with plots indicating abundance in relation to the size of the intergenic region. **d** Transcript abundances for Gene 2. For **c** and **d**, the heat score (count) indicates the number of genes with both the same PTM or transcript abundance and the same size intergenic region. MNase, micrococcal nuclease; kb, kilobases; bp, base pairs

(Fig. 6c, measured at the TSS of the second gene of the gene pair, see schema in Fig. 6a). This difference appeared to be due to the overlap of the two divergent TSS regions as the signal markedly increased for the intergenics of divergent gene pairs that were shorter than 1 kbp (Fig. 6c). Consistent with this observation, no significant difference was detected between the median expression levels of divergent and tandem genes (Fig. 6d). Note that transcript abundances were not correlated for the two genes of a divergent gene pair (Pearson coefficients were 0.042 for all the divergent pairs and 0.054 for divergent pairs closer than 600 bp; Additional file 4: Figure S8), despite the presence of a shared chromatin domain at the TSSs and correlated histone PTM signals immediately downstream of each TSS when divergent genes were closer than 600 bp (Additional file 4: Figure S8).

### Histone PTM patterns during the *Ectocarpus* life cycle

To relate patterns of histone modification to changes in gene expression during the *Ectocarpus* life cycle, ChIP-seq analysis was used to compare the distributions of histone PTMs during the sporophyte and gametophyte generations. Overall, the distribution of PTMs was stable between the two life cycle generations. For example, the six PTMs associated with actively expressed genes, H3K4me2, H3K4me3, H3K9ac, H3K14ac, H3K27ac and H3K36me3, were either stably present or stably absent for between 82.7 and 97.1% of the 17,406 genes genome-wide, depending on the PTM. This analysis indicated that there were not any major, genome-wide changes in the patterns of histone PTMs associated with alternation between life cycle generations. To focus more specifically on the changes in the patterns of histone PTMs associated with life cycle-related changes in the gene expression, we analysed the presence and absence of marks at genes that were differentially regulated between the two life cycle generations (Fig. 7a).

A comparison of gene expression patterns in the sporophyte and gametophyte, based on RNA-seq data generated using the same biological samples as were used for the ChIP-seq analysis, identified 774 genes that exhibited a generation-biased pattern of expression (padj < 0.05, fold change > 2, TPM > 1; Fig. 7a, Additional file 5: Table S5). We will refer to these 774 genes as generation-biased genes (GBGs). Analysis of the predicted functions of the GBGs using a system of manually assigned functional categories [7] indicated significant enrichment in several functional categories, in particular, in cell wall and extracellular proteins and proteins of unknown function ($\chi^2$ test padj 3.73e$^{-40}$ and 1.56e$^{-5}$, respectively; Table S6). Prediction of subcellular location using Hectar [24], indicated that the GBG protein set was also significantly enriched in secreted proteins ($\chi^2$ test, 2.20e$^{-27}$; Table S7). These observations are consistent with a recent analysis using an independently identified set of GBGs [7] and suggest possible important roles for the cell wall and intercellular interactions in the specific functions of the two generations of the life cycle. Interestingly, this set of differentially expressed genes was significantly enriched in long non-coding RNAs (lncRNAs; Additional file 5: Table S8; $\chi^2$ test, 1.52e$^{-15}$), and 19 of the 72 differentially expressed lncRNAs were adjacent to a differentially expressed protein-coding GBG (including four lncRNA/protein-coding gene pairs with overlapping transcripts; Additional file 5: Table S8). In most cases (84%), the lncRNA and its adjacent protein-coding gene were co-ordinately upregulated

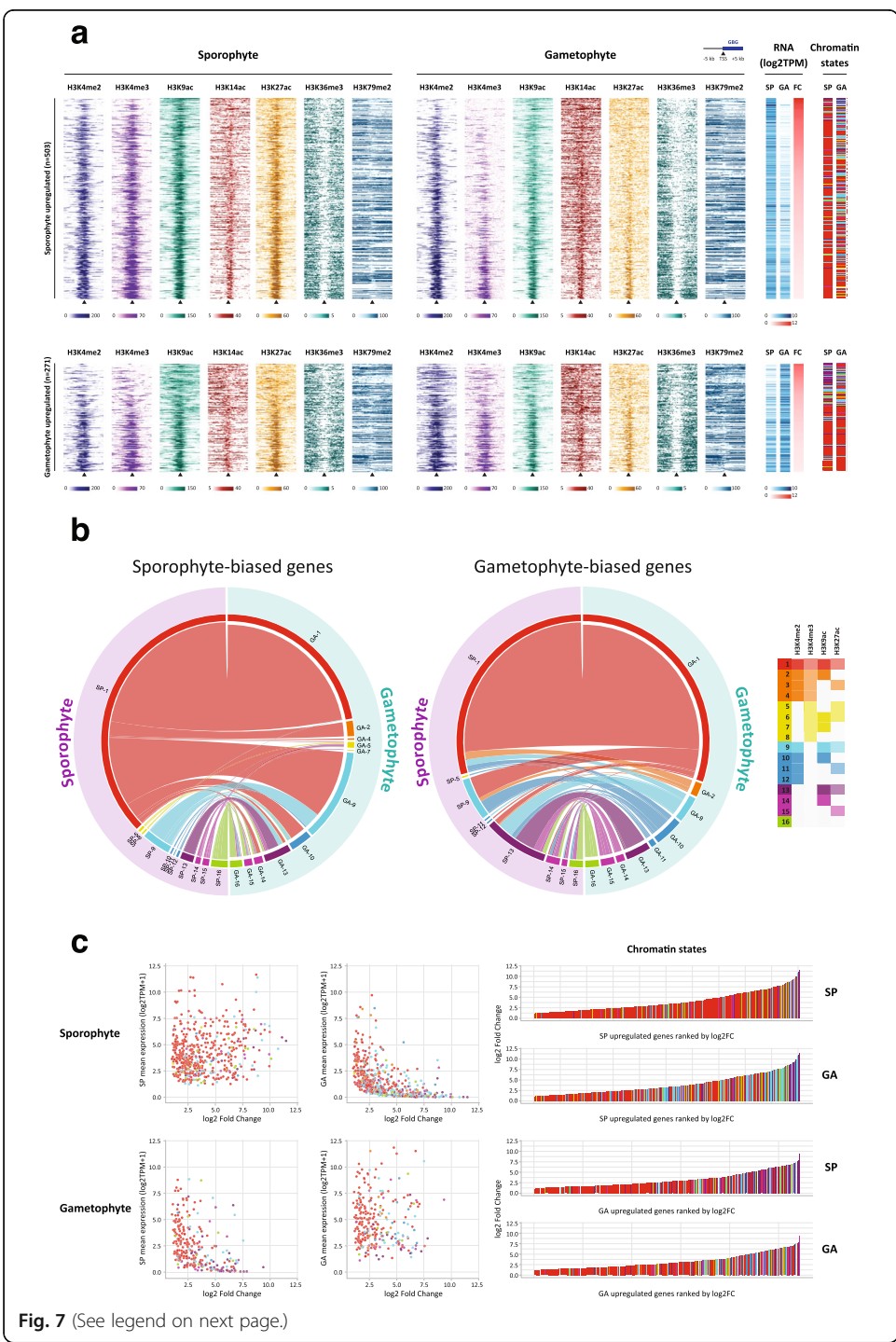

**Fig. 7** (See legend on next page.)

(See figure on previous page.)

**Fig. 7** Alterations of histone PTM patterns at generation-biased genes during the two generations of the life cycle. **a** Patterns of histone PTMs over 10 kb regions surrounding the TSSs of sporophyte-biased (upper panel) and gametophyte-biased (lower panel) genes during the sporophyte (left) and gametophyte (middle) generations. The heatmaps on the right show transcript abundance during the two generations and log2 fold change between generations. **b** Circos plots comparing the chromatin states (combinations of histone PTMs) at the transcription start sites (TSS) of sporophyte-biased (left) and gametophyte-biased (left) genes in chromatin from the sporophyte (mauve) and gametophyte (green) generations of the life cycle. Colours correspond to chromatin states one to 16 as indicated in the key. **c** Plots of fold change against transcript abundance (TPM) for sporophyte- and gametophyte-biased genes during the two generations (left panel) along with ranked plots of fold change (right panel). Colour coding corresponds to chromatin states

during the same generation of the life cycle (Table S9). These observations suggest a possible role for lncRNAs in the regulation of adjacent genes in *Ectocarpus*.

To analyse changes in histone PTMs at GBGs between life cycle generations, 16 chromatin states were defined based on different combinations of four TSS-localised PTMs that exhibited clear patterns of presence and absence: H3K4me2, H3K4me3, H3K9ac and H3K27ac (Fig. 7b). In most cases, when GBGs changed chromatin state during the transition between the two generations, increased transcript abundance was correlated with the acquisition of additional activation-associated PTMs (76.1% of the GBGs that changed chromatin state). However, the majority of GBGs (61.1%) did not change chromatin state between generations, although we noted that most of these genes (81.4%) were marked with all four activation-associated PTMs during both generations. Taken together, these observations are consistent with a correlation between the presence of the four TSS-localised marks and gene activation but indicate that additional mechanisms may be necessary to induce differential gene expression. Therefore, if the histone PTMs have a role in regulating transcription, this role would most likely be to facilitate or potentiate transcription rather than directly activating gene transcription. Note, however, that it is also possible that some or all of the observed histone PTM modifications were a consequence, rather than a cause, of transcription.

We detected more sporophyte-biased (503) than gametophyte-biased (271) genes ($\chi^2$ test, $7.49e^{-17}$), and the former were significantly more strongly upregulated than the latter (4.2 and 2.8 mean log2TPM fold changes, respectively; Kruskal-Wallis rank sum test $p$ value = $4.0e^{-11}$). Overall, sporophyte-biased and gametophyte-biased genes showed similar patterns of transitions to new chromatin states (Spearman's rank correlation rho 0.61, $p$ value = 0.011). However, the sporophyte-biased genes exhibited a stronger tendency to switch to chromatin state 1 (the presence of all four activation-associated PTMs) than the gametophyte-biased genes (74.8% compared with 26.4% of chromatin state transitions, respectively; $z$ score 7.7857, $p < 0.00001$). This observation is consistent with the larger fold changes in transcript abundance (TPM) observed for sporophyte-biased genes as transitions to chromatin state 1 were associated with significantly larger fold changes than transitions to other states (Wilcoxon test, $p$ value < 0.0078 for all GBGs). We also noted that the GBGs that exhibited the highest fold changes also exhibited a stronger tendency to change chromatin state between generations (Fig. 7c). This was true for both sporophyte-biased and gametophyte-biased genes.

#### *Ectocarpus* lacks polycomb repressive complexes

Polycomb repressive complexes have been shown to play important roles in the regulation of a broad range of developmental processes in both animals and land plants [25, 26], and a key role has been proposed for Polycomb repressive complex 2 (PRC2) in the regulation of life cycle transitions in both *Arabidopsis* [10, 12–15] and the moss *Physcomitrella patens* [16, 17]. However, in contrast to the conservation and functional importance of these complexes in the land plant and animal lineages, a homology search failed to identify the core proteins of PRC2 in the *Ectocarpus* genome, and a similar result was obtained for PRC1 ([2]; Fig. 8, Additional file 5: Table S10).

This analysis did identify an orthologue of the WD domain protein RbAp48, but this protein is also known to be a key component of other regulatory complexes in the cell, such as CAF-1, NuRD and NURF [28]. Moreover, analysis of the *Cladosiphon okamuranus* [29], *Nemacystus decipiens* [30] and *Saccharina japonica* [31] genomes indicated that they too lack all the PRC2 and PRC1 core proteins except RbAp48, which is consistent with the absence of PRC complexes being a general feature of the brown algae (Table. S10). We cannot rule out the possibility that the brown algae possess highly diverged versions of the PRC2 and PRC1 complexes, but this would seem unlikely as the evolutionary time separating brown algae from plants and animals is similar to that separating the latter two lineages. Moreover, orthologues of all the PRC2 and PRC1 core components were found in the genome of the diatom *Phaeodactylum tricornutum*, which also belongs to the stramenopile lineage ([32]; Additional file 5: Table. S10). Analysis of the limited number of complete genomes available for the stramenopiles indicated that PRC1 was lost after divergence from the diatoms and PRC2 at a later stage after divergence from the Pelagophyceae (Fig. 8b). The absence of PRC2 in *Ectocarpus* was supported by the fact that the mass spectrometry analysis did not detect any evidence of tri-methylation of H3K27. Histone immunoblots with an antibody raised against H3K27me3 (Additional file 5: Table S11) weakly detected a protein of the expected size but the signal was only detected after a lengthy exposure (see legend to Additional file 4: Figure S3), and when this antibody was employed in a ChIP-seq experiment, we did not detect any specific chromatin immunoprecipitation (data not shown). Similarly, an antibody raised against H2AK119ub did not detect this histone PTM in *Ectocarpus* chromatin (Additional file 3: Figure S9). Therefore, taken together, these analyses indicated that *Ectocarpus* lacks the PRC2 and PRC1 polycomb complexes and, consequently, the histone modifications mediated by these complexes, tri-methylation of H3K27 and mono-ubiquitination of H2AK119.

### Discussion

#### Histone post-translational modifications in a multicellular brown alga

Brown algae are the third most developmentally complex phylogenetic group on the planet and include members whose body plans rival those of land plants in their cellular and developmental complexity. However, compared to animals and land plants, many aspects of brown algal developmental biology are still very poorly understood, including epigenetic regulatory mechanisms. The objective of this study was to provide a comprehensive overview of histone PTMs in the model brown alga *Ectocarpus* and to investigate the relationship between patterns of key PTMs across the genome and the

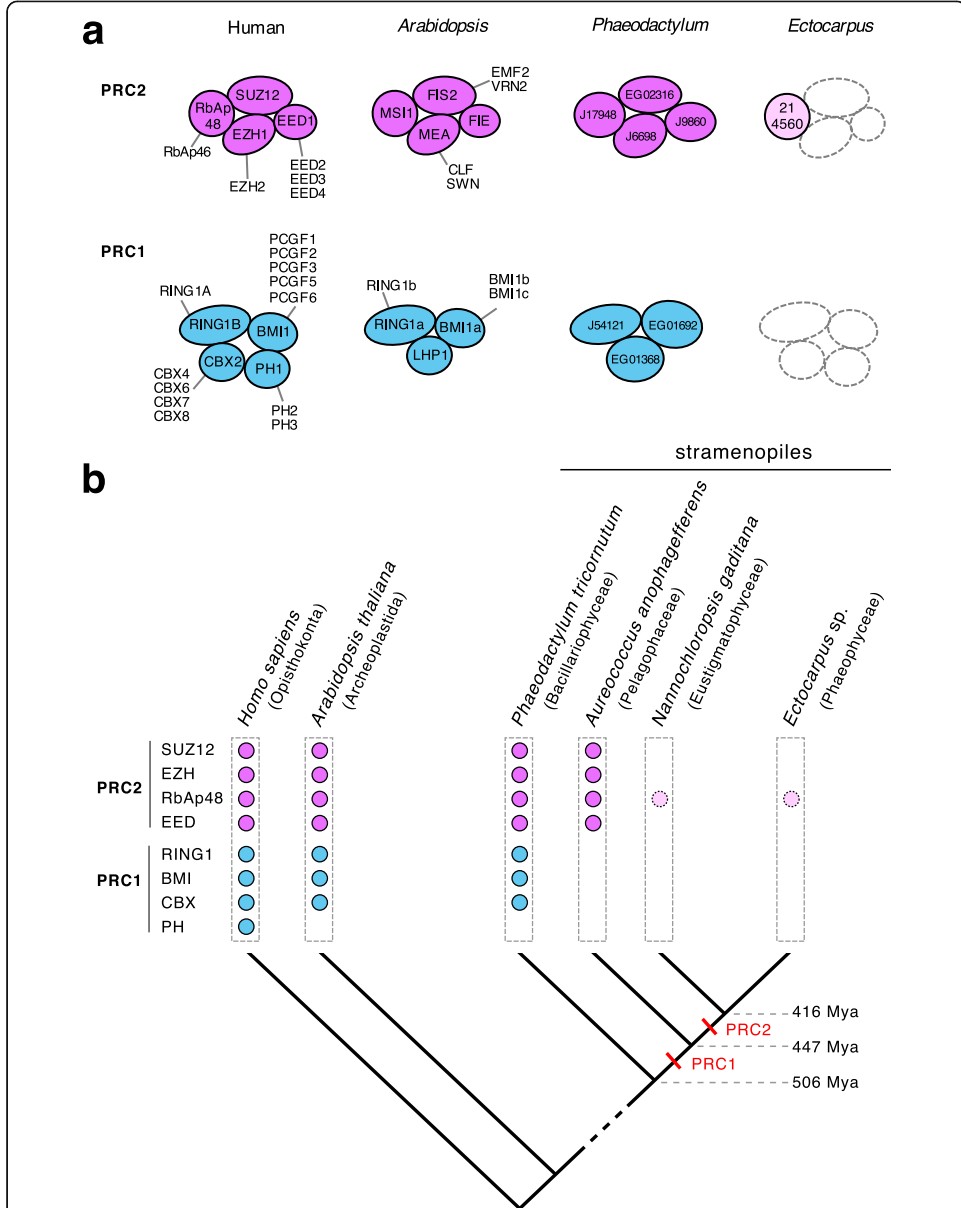

**Fig. 8** Loss of polycomb complexes from brown algae. **a** Core components of the polycomb complexes PRC2 (mauve) and PRC1 (blue) in *Homo sapiens*, *Arabidopsis thaliana*, *Phaeodactylum tricornutum* and *Ectocarpus* sp.. Mauve or blue circles indicate the presence of a PRC2 or PRC1 gene, respectively. RbAp48 is indicated in pink when it probably represents a component of other cellular complexes (see text for details). Alternative proteins are shown for each component. Empty, dotted circles indicate components that are absent from *Ectocarpus*. All *P. tricornutum* and *Ectocarpus* sp. protein names should be prefixed with Phatr3 or Ec-, respectively, e.g. Phatr3_J17948, Ec-21_004560. **b** Schematic phylogenetic tree indicating PRC2 and PRC1 genes present in *Homo sapiens*, *Arabidopsis thaliana* and four selected stramenopiles. Red bars indicate predicted approximate time points for loss of the PRC1 and PRC2 complexes during stramenopile evolution. Approximate divergence times (± 100 My) are based on Brown and Sorhannus [27]

developmental programmes that mediate alternation between the sporophyte and gametophyte generations during this seaweed's life cycle.

Mass spectrometry analysis of *Ectocarpus* histone preparations demonstrated that brown algal histones are subject to a broad range of PTMs. Most of the modifications detected had been reported previously for histones of organisms from other eukaryotic

supergroups, such as the land plants within the Archaeplastida or animals and fungi from the Opisthokonta. However, the analysis also confirmed some stramenopile-specific histone PTMs such as H2A.ZK9ac and H2A.ZK15ac and detected a previously unreported (and therefore possibly brown algal-specific) modification, H2A.ZR38me1. Overall, however, this study confirmed the conclusion, based on an analysis of diatom histone PTMs [22], that a large number of histone PTMs detected in other eukaryotic supergroups are conserved in the stramenopiles. This conclusion is consistent with the observation that histone molecules have been highly conserved during eukaryotic evolution [33, 34] and with evidence that many of the proteins involved in post-translational modification of histones can be traced back to the last eukaryotic common ancestor [35]. Indeed, genes encoding histone-modifying enzymes with highly conserved domain structures and domain sequences, such as the histone acetyltransferases, were detected in the *Ectocarpus* genome. For other families, it was difficult to assign predicted proteins to specific functions due to sequence divergence. Experimental analysis of protein function will therefore be necessary to identify the specific writers and readers of many histone PTMs in brown algae.

The *Ectocarpus* genome does not appear to encode the core components of the polycomb complexes PRC1 and PRC2. The absence of polycomb complexes in a complex multicellular organism may seem surprising from the perspective of animal and land plant model systems, but the components of PRC2 are known to exhibit a patchy distribution across the diverse eukaryotic supergroups [36], indicating that they have been repeatedly lost over evolutionary time. In this respect, the absence of polycomb complexes in brown algae represents a striking example of how comparative analysis of groups that have evolved complex multicellularity independently of plants and animals can be used to test the degree to which specific genetic systems (in this case polycomb regulation) have been essential for the evolution of complex multicellularity.

Analysis of ChIP-seq data allowed inferences to be made about the possible functional roles of eight histone PTMs in brown algae. H3K4me2, H3K4me3, H3K9ac, H3K14ac and H3K27ac were all detected predominantly at TSSs, and the degree to which genes were marked with these PTMs was proportional to their level of expression (transcript abundance), suggesting a role in promoting or facilitating gene transcription. Note, however, that we cannot exclude the opposite hypothesis that deposition of these PTMs is promoted by gene transcription.

Further evidence for a correlation between gene expression and the presence of the five TSS PTMs was provided by a comparative analysis of chromatin states at GBGs between life cycle generations, which indicated a broad correlation between accumulation of H3K4me2, H3K4me3, H3K9ac and H3K27ac at the TSS and increased expression of GBGs during the life cycle. However, patterns of histone PTMs at TSSs were not strictly correlated with gene expression indicating that, if these marks do facilitate transcription, they are not sufficient alone to induce gene expression. This conclusion was supported by a genome-wide analysis of closely spaced (< 600 bp) pairs of divergently transcribed genes, which showed that the TSS regions of gene pairs overlapped but the two genes were nonetheless independently regulated. At present, it is not clear what processes might be necessary, in addition to the deposition of histone PTMs at the TSS, to induce gene expression in *Ectocarpus*, but these could include, for example, the presence of specific transcription factors and the transcription machinery or the

presence of additional histone PTMs, including alternative acyl marks such as propionylation, butyrylation and crotonylation which have been shown to stimulate transcription in cell-free assays [37].

Peaks of H3K4me2, H3K4me3, H3K9ac, H3K14ac and H3K27ac are detected near the TSSs of active genes in both animals and land plants, and all of these marks have been associated with active gene expression in both lineages ([38–49], with the possible exception of H3K4me2, which may represent a repressive mark on plants [49, 50]). Overall, therefore, these five histone PTMs appear to have similar roles associated with gene activation at promoters in animals, land plants and brown algae, indicating that at least part of the "histone code" may be universal and therefore have a very deep evolutionary origin.

H3K36me3 marked gene bodies and was absent from TSSs and TESs, but its presence was also positively correlated with the level of gene expression. H3K36me3 also marks the bodies of active genes in both animals and land plants, although its distribution along the gene body differs in the two lineages. In *Arabidopsis* and rice, a strong peak was observed at the 5′ end of the gene [45, 48, 51], whereas strong signals have been reported at the 3′ ends [52] and over exons [53] of animal genes. H3K36me3 is catalysed by SETD2, which is associated with Pol II and is therefore directly linked to transcript elongation [54]. H3K36me3 has also been shown to be involved in alternative splicing in animals and plants [55, 56]. The position of H3K36me3 within *Ectocarpus* genes and the positive correlation with transcript abundance are consistent with an evolutionarily conserved role associated with transcript elongation.

H4K20me3 was associated with transposons and repeat sequences, both between and within genes. The presence of this mark at intronic transposons was inversely correlated with levels of gene expression, but its effect on gene expression may be indirect, for example, as a consequence of silencing intronic transposons. The distribution of H4K20me3 suggests a similar role to that observed in animals, where H4K20me3 is enriched at transposons and in heterochromatin [57–59] and where it has been shown to repress transposons [59]. Note that H4K20me3 appears to have a different role in land plants as it localises to euchromatin and is associated with transcriptional activation [60].

Heterochromatin is marked by methylation of lysine 9 of histone H3 in animals and land plants [61]. H3K9me3 is the most abundant heterochromatin marker in animals [62, 63]. In plants, heterochromatin is silenced by a feedback loop involving H3K9me2 and DNA methylation [64], whereas H3K9me3 is associated with euchromatin and appears to have a different function [51]. H3K9me2 and H3K9me3 were detected in *Ectocarpus* but appear to be present at low abundance, and ChIP-seq profiling did not provide any exploitable information about the distribution of these PTMs (unpublished results, data available at NCBI Gene Expression Omnibus accession GSE146369). More sensitive analysis methods will be necessary to investigate the roles of H3K9me2 and H3K9me3 in brown algae, but it is possible that other PTMs, such as H4K20me3 and H3K79me3, carry out at least some of the functions that have been attributed to these PTMs in other lineages.

H3K79me2 exhibited an unusual distribution pattern, marking discrete regions of the genome that often spanned multiple genes. The function of H3K79me2 is not clear, although H3K79me2-marked regions were strongly correlated with the presence of H4K20me3 and commonly included transposons (88.6% of H3K79me2 regions of > 5 kbp included a ≥ 400-bp repeated element). As observed for H4K20me3, the presence of

H3K79me2 was correlated with the repression of gene expression (lower levels of transcript abundance). Interestingly, genes marked with both H3K79me2 and H4K20me3 were expressed at significantly lower levels than genes marked with only one of the two PTMs, indicating that the mechanisms linking H3K79me2 and H4K20me3 with gene repression act independently.

Taken together, the above observations would be consistent with a role for H3K79me2 in the silencing of repeated elements (based on a similar argument to that proposed above for H4K20me3). If H3K79me2 does have a role in the silencing of transposons, it is not clear why the H3K79me2 regions often cover much larger regions than the transposons itself. One possible explanation may be that the regions extend to neighbouring TSSs in order to regulate transcriptional activity in a genomic region containing a transposon. We also noted that H3K79me2 tended to mark convergent gene pairs, even in regions that contained little repeated sequence. A possible additional role of H3K79me2 at these loci may be to regulate transcription in some manner to limit or respond to the formation of double-stranded RNA due to transcript overlap. Such double-stranded RNA could be problematic within the cell if it activates small RNA pathways, but note that an earlier analysis did not find any evidence that convergent gene pairs were a preferential source of sRNAs in *Ectocarpus* [65].

The distribution pattern observed for H3K79me2 in *Ectocarpus* is very different from the pattern observed in animals, where H3K79me2 marks the 5′ part of the gene body as part of a mutually exclusive pattern with H3K36me3, which marks the 3′ part [53, 66, 67]. The H3K79me2/H3K36me3 boundary corresponds to the first internal exon [53]. In animals, both H3K79me2 and H3K36me3 are considered to be associated with active transcription [66–68], and more specifically transcript elongation, although conflicting evidence has been reported [52]. In addition, a role has been proposed for H3K79me2 in the regulation of alternative splicing [69], although again there is evidence to the contrary [53]. In fungi, H3K79me2 has also been found to be associated with euchromatin and to be excluded from heterochromatin [70]. H3K79 methylation has not been detected in land plants [21]. Based on the different distribution patterns, H3K79me2 would appear to have different roles in animals and in brown algae.

We also observed that a region of chromosome 6, containing a large, inserted DNA virus genome, was extensively marked with both H3K79me2 and H4K20me3. The viral genes at this locus have been shown to be transcriptionally silent in a range of tissues and life cycle stages [18], suggesting possible roles for H3K79me2 and H4K20me3 in defending the alga against lysogenic viral infections by silencing inserted viral genomes.

### Histone PTM patterns during the *Ectocarpus* life cycle

To our knowledge, this is the first study to have compared patterns of histone PTMs across the two generations of a haploid-diploid life cycle (although previous studies have looked at PTMs associated with the repression of germline genes during the sporophyte generation [71, 72]). Overall, genome-wide patterns of histone PTM were found to be stable between the *Ectocarpus* sporophyte and gametophyte generations, with the marks deposited at most genes remaining unchanged. The *Ectocarpus* sporophyte and gametophyte are very similar morphologically (near-isomorphy), and it is possible that more marked changes in histone PTM patterns occur during the life

cycles of brown algal species (such as kelps) that exhibit greater differences between the sporophyte and gametophyte generations. However, note that there does not appear to be a strong correlation between the degree of morphological heteromorphy between life cycle generations and the proportion of GBGs in the genome [73]. Analysis of the epigenetic status of GBGs during the life cycle indicated that the differential expression patterns of these genes were correlated with modifications of multiple TSS-localised histone PTMs. Again, as stated above, is not clear at present whether these histone PTM modifications are involved in facilitating or potentiating gene upregulation or whether, on the contrary, changes in the transcription status of these genes lead to changes in the pattern of histone PTMs at the TSS.

## Conclusion

This study provides a first overview of the nature and functions of histone PTMs in the brown algae, a group that has evolved complex multicellularity independently of the animal and land plant lineages. The general, emerging picture is that epigenetic regulatory mechanisms are broadly conserved with those of other eukaryotic supergroups with some key differences that will potentially provide important insights into the epigenetic regulation of developmental processes, including novel histone modifications, unusual patterns of deposition of known PTMs such as H3K79me2 and the absence of polycomb complexes. The data and resources generated by this study will provide a foundation for future studies aimed at understanding epigenetic processes in the brown algae.

## Material and methods

### Strains and growth conditions

The *Ectocarpus* sp. strain used in this study was Ec32 [74]. *Ectocarpus* was cultivated as described previously [75]. Cultures were grown at 13 °C with a 12-h/12-h day/night cycle and 20 μmol photons $m^{-2}$ $s^{-1}$ irradiance. Gametophytes were obtained from germinating meio-spores; the sporophyte generation corresponded to parthenosporophytes derived from parthenogenetic gamete development.

### Detection of histone PTMs using mass spectrometry

*Ectocarpus* histone proteins were isolated using the method described by Tirichine et al. [76]. Briefly, histones were extracted from isolated nuclei in 1 M $CaCl_2$, 20 mM Tris HCl pH 7.4 in the presence of a cocktail of protease inhibitors. After precipitation of the acid-insoluble fraction in 0.3 M HCl, the histones were precipitated by dropwise addition of TCA, centrifuged and the pellet washed with 20% TCA and 0.2% HCl.

Gel purification and digestion of histones and mass spectrometry analysis were carried out essentially as described by Veluchamy et al. [22]. Briefly, histone proteins, excised from a 14% SDS-polyacryamide gel, were digested overnight with endoproteinase (12.5 ng/μl), trypsin (Promega), chymotrypsin (12.5 ng/μl, Promega), ArgC (12.5 ng/μl, Promega) or elastase (20 ng/μl Sigma-Aldrich). Peptides were analysed using a Q Exactive HF-X (Thermo Scientific) or a TripleTOF™ 6600 (ABSciex) mass spectrometer coupled to a RSLCnano system (Ultimate 3000, Thermo Scientific, Bremen, Germany). Spectra were generated using the Xcalibur (version 2.0.7) and analysed with Mascot™

(version 1.4, Thermo Scientific) using an in-house database consisting of the complete *Ectocarpus* proteome available at Orcae [77]. The mass spectrometry proteomics data have been deposited in the PRIDE database [78] via ProteomeXchange with identifier PXD013535 [79]. Details of the liquid chromatography-tandem mass spectrometry parameters are provided in the Supplementary Methods.

### Detection of histone PTMs using immunoblots

Commercially available antibodies (Additional file 5: Table S11) for a broad range of histone PTMs were tested against *Ectocarpus* chromatin preparations using immunoblots as described previously [80]. For antibodies that gave no signal or a weak signal with *Ectocarpus* extracts, human HeLa cell extracts were used as a positive control. Briefly, HeLa cells were harvested and centrifuged at room temperature at 1500 rpm for 5 min and then lysed in ice for 20 min using lysis buffer A (5 mM PIPES pH 8.0, 85 mM KCl, 0.5% IGEPAL® CA-630). This cell extract was used for the immunoblot. An anti-histone H4 antibody (Millipore 05-858) was used as a loading control.

### Genome-wide detection of histone PTMs

The ChIP-seq experiments were carried out in two batches, the first for H3K4me3, H3K9ac, and H3K27ac and the second for H3K4me2, H3K14ac, H3K36me3, H4K20me3, H3K79me2 and H3K27me3. RNA-seq data (see below) was generated for both batches to ensure that the histone PTM and gene expression data were fully compatible. For ChIP-seq, *Ectocarpus* tissue was fixed for 5 min in seawater containing 1% formaldehyde, and the formaldehyde eliminated by rapid filtering followed by incubation in PBS containing 400 mM glycine. The nuclei were isolated by grinding in liquid nitrogen and in a Tenbroeck Potter in nuclei isolation buffer (0.1% Triton X-100, 125 mM sorbitol, 20 mM potassium citrate, 30 mM MgCl2, 5 mM EDTA, 5 mM β-mercaptoethanol, 55 mM HEPES at pH 7.5 with cOmplete ULTRA protease inhibitors), filtering through Miracloth and then washing the precipitated nuclei in nuclei isolation buffer with, and then without, Triton X-100. Chromatin was fragmented by sonicating the purified nuclei in nuclei lysis buffer (10 mM EDTA, 1% SDS, 50 mM Tris-HCl at pH 8 with cOmplete ULTRA protease inhibitors) in a Covaris M220 Focused-ultrasonicator (duty 25%, peak power 75, cycles/burst 200, duration 900 s at 6 °C). The chromatin was incubated with an anti-histone PTM antibody (Additional file 5: Table S11) overnight at 4 °C and the immunoprecipitation carried out using Dynabeads protein A and Dynabeads protein G. Following immunoprecipitation and washing, a reverse cross-linking step was carried out by incubating for at least 6 h at 65 °C in 200 mM NaCl, and the samples were then digested with proteinase K and RNAse A. Purified DNA was analysed on an Illumina HiSeq 4000 platform with a single-end sequencing primer over 50 cycles. At least 20 million reads were generated for each immunoprecipitation (Additional file 5: Table S12).

Nuclei were prepared for MNase digestion using the same procedure as that used for ChIP samples except that the isolation buffer did not contain EDTA. Each sample of nuclei was digested with 400 agarose gel units of MNase for 10 min at 37 °C in MNase reaction buffer (5 mM CaCl$_2$, 60 mM KCl, 0.5 mM DTT, 15 mM NaCl, 125 mM sorbitol, 50 mM Tris-HCl pH 7.5). Following lysis of the nuclei, reverse cross-linking

and digestion with proteinase K and RNAse A, 120–210-bp fragments were excised from a 2% agarose gel and sequenced on a HiSeq 4000 platform (Additional file 5: Table S12).

The ChIP-seq and MNase digestion datasets have been deposited in the NCBI Gene Expression Omnibus database, along with the associated RNA-seq data, under the accession number GSE146369.

Quality control of the sequence data was carried out using FASTQC [81]. Poor quality sequence was removed, and the high-quality sequence was trimmed with Flexcat [82]. Illumina reads were mapped onto the *Ectocarpus* genome ([18]; available at Orcae [77]) using Bowtie [83]. Peaks corresponding to the regions enriched in PTMs were identified using the MACS2 (version 2.1.1) callpeak module (minimum FDR of 0.01) and refined with the MACS2 bdgpeakcall and bdgbroadcall modules [84]. Additional file 5: Table S13 summarises the correspondence between PTM peaks and genes across the *Ectocarpus* genome. Co-localised peaks corresponding to regions enriched in several PTMs were detected using MACS2 outputs in BedTools multiinter [85]. Overlaps of PTM peaks with other genomic features were analysed using BedTools intercept [85]. The analysis of the association of H3K79me2 domain borders with TSSs and TESs was carried out in R, using the function gr.rand from the package gUtils to generate the random anchors. Heatmaps, average tag graphs and coverage tracks were plotted using the EaSeq software [86], deepTools [87] or pyGenomeTracks (https://pypi.org/project/pyGenomeTracks/). Circos graphs were generated using the Circos software [88]. These analyses were carried out for two biological replicates for each PTM during both the sporophyte and gametophyte generations of the life cycle. Pearson correlation analysis between replicates was performed with deepTools 3.2.0 [87]. Replicate samples were strongly correlated (Pearson correlations of 0.87 to 0.99; Additional file 4: Figure S10). Normalisation was carried out using the input chromatin data (Additional file 5: Table S12). Genome-wide analyses (e.g. Fig. 2) used data from sporophytes unless stated otherwise.

### Comparisons of sporophyte and gametophyte transcriptomes using RNA-seq

RNA for transcriptome analysis was extracted from the same duplicate sporophyte and gametophyte cultures that were analysed for each of the two ChIP-seq experiments using the Qiagen RNeasy plant mini kit with an on-column deoxyribonuclease I treatment. RNA quality and concentration were then analysed on a Qubit® 2.0 Fluorometer using the Qubit RNA BR assay kit (Invitrogen, Life Technologies, Carlsbad, CA, USA). Between 49 and 107 million sequence reads were generated for each sample on an Illumina HiSeq 4000 platform with a single-end sequencing primer over 150 cycles (Additional file 5: Table S12). Quality trimming of raw reads was carried out with Flexcat [82], and reads of less than 50 nucleotides after trimming were removed. Tophat2 [89] was used to map reads to the *Ectocarpus* genome [18], and the mapped sequencing data was processed with featureCounts [90] to obtain counts for sequencing reads mapped to exons. Expression values were represented as TPM. Differential expression was detected using the DESeq2 package (Bioconductor [91]) using an adjusted *p* value cut-off of 0.05 and a minimal fold change of 2. The set of GBGs corresponded to genes

that were significantly differentially regulated between life cycle generation in both of the ChIP-seq experiments.

### Searches for histone and histone-modifying enzyme encoding genes in *Ectocarpus*

Histone and histone-modifier genes were detected in the *Ectocarpus* genome [18] using Blast [92] and manually reannotated when necessary, using GenomeView for structural reannotation [93].

## Supplementary Information

---

**Additional file 1.** Supplementary methods.

**Additional file 2: Figure S1.** Histone gene clusters in the *Ectocarpus* genome.

**Additional file 3: Figure S2.** Representative MS/MS spectra showing the identification of histone modifications.

**Additional file 4: Figure S3.** Immunoblots of histone PTMs. **Figure S4.** Genomic distribution of the histone PTMs H3K79me2 and H4K20me3. **Figure S5.** Boundaries of H3K79me2 regions are preferentially located near TSSs and TESs. **Figure S6.** Comparison of genes marked with H3K79me2 alone, H4K20me3 alone or both H3K79me2 and H4K20me3. **Figure S7.** H3K79me2, H4K20me3, H3K36me3, H3K4me2, H3K4me3, H3K9ac, H3K14ac and H3K27ac signals for a region of chromosome 6 spanning an inserted viral genome. **Figure S8.** Correlations of transcript abundances and histone PTM signals for divergently transcribed pairs of genes. **Figure S9.** The histone PTM H2AK119ub1 was not detected in *Ectocarpus*. **Figure S10.** Pearson correlation scores for comparisons of the genomic distributions of ChIP-seq signal peaks for duplicate assays of the eight histone PTMs during both the sporophyte and gametophyte generations.

**Additional file 5: Table S1.** Histone proteins in *Ectocarpus*. **Table S4.** H4K20me3 peaks associated with different transposon families. **Table S5.** Genes with significantly different transcript abundances during the gametophyte and sporophyte generations. **Table S6.** Identification of enriched functional categories in the proteins encoded by the generation-biased gene set. **Table S7.** The generation-biased gene set is enriched in secreted proteins. **Table S8.** Analysis of long non-coding RNA genes that were differentially expressed during the gametophyte and sporophyte generations. **Table S9.** lncRNA GBGs with an adjacent protein-coding GBG. **Table S10.** Presence and absence of polycomb complex proteins in animals, land plants and stramenopiles. **Table S11.** Anti-histone-PTM antibodies used in this study. **Table S12.** Sequence data generated by this study. **Table S13.** Presence or absence of histone PTMs and RNA-seq TPMs for all *Ectocarpus* genes in the sporophyte and the gametophyte generations.

**Additional file 6: Table S2.** Putative PTM writers and erasers in *Ectocarpus* sp.

**Additional file 7: Table S3.** Presence or absence of histone post-translational modifications in seven species from diverse eukaryotic supergroups.

**Additional file 8.** Review history.

---

#### Acknowledgements

We thank the Institut Français de Bioinformatique and the Roscoff Analysis and Bioinformatics for Marine Science platform ABiMS (http://abims.sb-roscoff.fr) for providing computing and data storage resources and Houda Benhelli-Mokrani for providing us with HeLa cells.

#### Peer review information

#### Review history

The review history is available as Additional file 8.

#### Authors' contributions

SB prepared the histones for mass spectrometry analysis and developed the ChIP-seq protocol with help from LT. SB carried out the ChIP-seq analysis. BL and DL carried out the mass spectrometry analysis of histone PTMs and analysed the data from these experiments. YW, SB and LT carried out the immunoblots. SB and JMC analysed the ChIP-seq data with input from LT and SMC. DS carried out the additional experimental work. SB and JMC wrote the manuscript with input from the other authors. All authors read and approved the final manuscript.

#### Funding

This work was supported by the Centre National de la Recherche Scientifique, Sorbonne University (including a PhD grant for SB), the Agence Nationale de la Recherche projects Epicycle and Idealg (ANR-19-CE20-0028-01 and ANR-10-BTBR-04-01, respectively) and the European Research Council (grant agreement 638240). DL acknowledges support from the Région Ile-de-France and the Fondation pour la Recherche Médicale.

## Availability of data and materials
All sequence data used in this manuscript (Additional file 5: Table S12) have been deposited in the NCBI Gene Expression Omnibus (https://www.ncbi.nlm.nih.gov/geo/) under the accession number GSE146369 [94]. The *Ectocarpus* genome assembly is available through Orcae (https://bioinformatics.psb.ugent.be/orcae/overview/EctsiV2). The mass spectrometry proteomics data is available in the PRIDE database under accession number PXD013535 [79].

## Ethics approval and consent to participate
Not applicable.

## Consent for publication
Not applicable.

## Competing interests
The authors declare that they have no competing interests.

## Author details
[1]CNRS, Sorbonne Université, UPMC University Paris 06, Algal Genetics Group, UMR 8227, Integrative Biology of Marine Models, Station Biologique de Roscoff, CS 90074, F-29688 Roscoff, France. [2]Université de Nantes, CNRS, UFIP, UMR 6286, F-44000 Nantes, France. [3]Institut Curie, PSL Research University, Centre de Recherche, Laboratoire de Spectrométrie de Masse Protéomique, 26 rue d'Ulm, 75248 Paris, Cedex 05, France. [4]Current address: Max Planck Institute for Developmental Biology, Max-Planck-Ring 5, 72076 Tübingen, Germany.

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

## 

