## [**Additional file 8.** Review history. · Genome Biology]

Review History

First round of review

Reviewer 1

Are you able to assess all statistics in the manuscript, including the appropriateness of statistical tests used? No, I do not feel adequately qualified to assess the statistics.

Comments to author:

This is an interesting first report of the "histone code" in a neglected group of organisms, the brown algae. The Author find broad conservation of marks associated with expressed genes with association of H3K4me3 and H3/H4 acetylation with the promoter, while H3K36me3 is associated with elongation and present over gene bodies as in many other eukaryotes. The striking difference with most other eukaryotes is the lack of H3K27me3 deposited by PRC2. The authors explain this absence by the lack of PRC2. This is apparently derived as PRC2 is present in more "basal" brown algae. This report is important for the community of researchers interested in genomes and chromatin because it fills a gap in our knowledge of evolution of the tool kit that regulates gene expression.

The work is properly executed using standard methods Mass spec and ChIPseq together with MNase-seq.

There are a few aspects of the paper that should be improved

Major points:

1. Ubiquitination was searched using mass spec but not detected on H2A, which could indicate the lack of ubiquitination associated with PRC1. The Authors should attempt to solve this problem by using specific antibodies and provide Western blots
2. The study reports H3K9me2/3 detection using antibodies in Western blots but no report is made on this mark. A Western blot should be added. The lack of detection of K9me and K4me in mass spec data is common unless specific procedures are taken. So H3K9me2/3 exists in Ectocarpus and it should be profiled genome wide and reported here. This is probably the mark missing to address point #3.
3. Although there is the note "Almost all (94.6%) of the H4K20me3 peaks were associated with repeated sequences or transposons, predominantly in intergenic regions but also in introns." with a much longer commentary on that note in the Discussion, there are no results shown (except a vague mention of Fig.3). The report lacks investigation of the non coding part of the genome including repeats and transposable elements that have been annotated by the Authors in previous work. This should be added here, because the Authors must have the data and the report is incomplete without this. Especially because the abstract claims "The overview of the nature and functions of histone PTMs in the brown algae presented here will provide a foundation for future studies"
4. the title "epigenetic reprogramming during ectocarpus life cycle" is misleading. First none of the marks studied here are "epigenetic" marks (only K27me3 and K9me2/3 are defined as such when the term is used properly). Second what is shown here reflects the change of transcription program, but has nothing in common with epigenetic reprogramming such as described in mammals where an histone modifications or DNA methylation is erased and reinstalled. Here as quoted by the Authors "Overall, the distribution of PTMs was stable between the two life cycle generations." And "there were not any major, genome-wide changes in patterns of histone PTMs associated with alternation between life cycle generations.:" These two statements contradict the very notion of reprogramming and this term should not be used, because it implies very broad changes of patterns of marks triggered by

events other than transcription. Please remove the term reprogramming and moderate the statement both in abstract, introduction, result and discussion.

Reviewer 2

Are you able to assess all statistics in the manuscript, including the appropriateness of statistical tests used? Yes, and I have assessed the statistics in my report.

Comments to author:

The manuscript by Bourdareau et al. reports on a series of genomics, proteomics and bioinformatics analyses of histone modifications in the brown alga *Ectocarpus*. They provide novel and convincing data on the presence and distribution of a multitude of likely histone modifications, including several new histone modifications determined through proteomics analyses of histone preparations from *Ectocarpus*. It is a shame that no antibodies exist against these 'new' histone modifications, such as H2A.ZK9ac or H2A.ZK20ac/me so that their relative amounts could be ascertained and their genome wide distribution be determined.

The authors did apply extensive ChIP-seq analyses to generate genome-wide maps of eight histone marks often assayed in many other eukaryotes. Five of these marks, H3K4me2, H3K4me3, H3K9ac, H3K14ac and H3K27ac were shown to be distributed within transcriptionally active regions of the *Ectocarpus* genome, and significantly depleted from the gene bodies while being enriched around the TSS. These observations, together with the observed genome-wide distribution of H3K36me3 mark, are in agreement with a rich body of literature across all eukaryotic lineages showing their association with transcriptionally active regions, and bring little novelty to chromatin modification distribution field, but may be of interest to comparative epigenomics field.

On the other hand, H4K20me3 mark was found associated with silent regions, mostly transposon-rich domains, but, interestingly in gene introns as well. The authors explain this by the fact that intron sequences comprise 40% of the *Ectocarpus* genome. I presume the authors' assumption there is that repeats and remnants of repeats are also likely to be found in many intron sequences? It'd be worth saying that outright, especially if it can be backed up by information from the *Ectocarpus* genome analysis that the authors should be quite familiar with. One recommendation I could make here would be to include a discussion on what roles might H3K9 methylation have in the silencing of these regions, in the absence of empirical data on H3K9me2/3 presence and distribution in *Ectocarpus*.

Interestingly, the authors show that H3K79me2 mark can be found in large genomic blocks of several kb, often spanning several genes, altogether spanning a combined 1/3 of the genome. Intriguingly, while the genes contained within the H3K79me2 blocks are less transcriptionally active and have an overall lower active gene marks (like H3K4me3 /H3K9ac / H3K27ac) the borders of these H3K79me2 domains seem to be defined by TSS of genes flanking the domain, implying that some sort of transcriptional interference with the establishment or spreading of the H3K79me2 mark. The authors make a nice comparison with the contrasting distribution of this mark in animals, and some of the ideas they presented in the discussion are nice food for thought. It is unfortunate that there is no genetics in *Ectocarpus* that would enable testing some of their models and hypotheses.

The authors also make a convincing argument using proteomic and genomics data that both PRC2 and PRC1 complexes as well as H3K27me3 mark are absent from *Ectocarpus*, and therefore conclude that Polycomb silencing is missing from *Ectocarpus* repertoire of gene silencing mechanisms.

The section on 'epigenetic reprogramming' during the *Ectocarpus* life cycle is probably the weakest

component of the manuscript, since the authors provide little evidence of any truly epigenetic reprogramming between the sporophyte and gametophyte. While they do identify 774 genes that exhibited generation-biased pattern of gene expression (including, interestingly, a significant enrichment of lncRNA in this set, something that could be analyzed in greater depth during revision, especially having in mind huge correlation in coordinated regulation of lncRNA and its adjacent coding gene), only a third of these genes show a change in chromatin state (as defined by a set of histone H3 modifications). While the causal relationship between gene expression and the chromatin marks analyzed is interpreted as 'histone-first' model ("epigenetics"), it is worth pointing out that the act of transcription itself may be what is recruiting these histone modifications (rather than the other way around), and therefore making the epigenetic argument very weak. I would recommend this section be rewritten and perhaps re-focused.

Overall, this manuscript reports on a series of ChIP-seq, RNA-seq and proteomics data that are descriptive in nature, and while well the experiments were well executed and contribute a large new body of data to the literature, it does not allow for any causal/functional inferences to be made. This is always an issue with working on orphan model organisms with limited genetic toolkit, and I would encourage the authors to revise their manuscript as indicated throughout this review in order to make it suitable for publication in *Genome Biology*.

Reviewer reports:

Please consider the following questions:

- Are the methods appropriate to the aims of the study, are they well described, and are necessary controls included? If not, please specify what is required.
- Are the conclusions adequately supported by the data shown? If not, please explain
- Are sufficient details provided to allow replication and comparison with related analyses that may have been performed? If not, please specify what is required.
- Does the work represent a significant advance over previously published studies?
- Is the paper of broad interest to others in the field, or of outstanding interest to a broad audience of biologists?

Reviewer #1:

This is an interesting first report of the "histone code" in a neglected group of organisms, the brown algae. The Author find broad conservation of marks associated with expressed genes with association of H3K4me3 and H3/H4 acetylation with the promoter, while H3K36me3 is associated with elongation and present over gene bodies as in many other eukaryotes. The striking difference with most other eukaryotes is the lack of H3K27me3 deposited by PRC2. The authors explain this absence by the lack of PRC2. This is apparently derived as PRC2 is present in more "basal" brown algae. This report is important for the community of researchers interested in genomes and chromatin because it fills a gap in our knowledge of evolution of the tool kit that regulates gene expression.

The work is properly executed using standard methods Mass spec and ChIPseq together with MNase-seq.

There are a few aspects of the paper that should be improved

Major points:

1. Ubiquitination was searched using mass spec but not detected on H2A, which could indicate the lack of ubiquitination associated with PRC1. The Authors should attempt to solve this problem by using specific antibodies and provide Western blots

*Reply: We have carried out a western blot with an anti-H2AK119ub1 antibody and show that, whilst it detects an antigen on human (HeLa) cell histones, there is no signal with an *Ectocarpus* histone preparation (figure S9).*

2. The study reports H3K9me2/3 detection using antibodies in Western blots but no report is made on this mark. A Western blot should be added. The lack of detection of K9me and

K4me in mass spec data is common unless specific procedures are taken. So H3K9me2/3 exists in *Ectocarpus* and it should be profiled genome wide and reported here. This is probably the mark missing to address point #3.

Reply: The revised manuscript now includes a figure showing detection of H3K9me2, H3K9me3, H3K4me2 and H3K4me3 PTMs by immunoblot (figure S3).

Concerning the lack of detection of histone H3K9me and H3K4me in mass spectrometry data. For H3K4me, the reviewer is correct that it is usually necessary to carry out a propionylation to be able to detect the small peptides methylated at this residue. However, the propionylation approach requires gel purification of histone H3 and, unfortunately, the gel resolution of the *Ectocarpus* samples was not sufficient to precisely localise and isolate H3. As an alternative approach, we tried modification of the % TFA and the use of a direct injection set-up without pre-concentration, to avoid loss of hydrophilic peptides. This latter approach has been used successfully to detect methyl, dimethyl and trimethyl modification of histone H3K4 in other species such as *Phaeodactylum tricornutum* but it did not detect H3K4me in *Ectocarpus*.

As far as histone K9me2/3 is concerned, we tested a large number of enzymes and two of the enzymes (Trypsin and ArgC) produced peptides that should have allowed detection of these modifications if they had been present at a concentration above the detection level (propionylation is not required for the detection of this PTM). Failure to detect these histone PTMs by mass spectrometry probably reflects their low abundance in *Ectocarpus* chromatin (i.e. the quantities of these modified peptides were presumably below the mass spectrometry detection limit). This conclusion is supported by the weak signal observed with K9me2/3 antibodies in immunoblots (figure S3).

ChIP-seq analysis of H3K9me2 and H3K9me3 was carried out in parallel with the other histone PTMs discussed in the manuscript. Very low amounts of DNA were recovered for these marks despite pooling of material from multiple immunoprecipitations for each replicate. We believe that the low yields were due to low abundance of these two histone PTMs in *Ectocarpus* chromatin, as indicated by the weak signals in the western blots (figure S3) and the absence of detection by mass spectrometry, although it is also possible that the PTMs are poorly recognised by the antibodies in the context of *Ectocarpus* histone H3 proteins. Libraries were nonetheless constructed and sequenced but, although extensive analyses of the data were carried out using several different strategies and programs, we were not convinced that the data contained any significant signal over background. We therefore decided not to include analysis of H3K9me2 and H3K9me3 in the manuscript. However, the ChIP-seq data for these two PTMs has been submitted to GEO along with the other data and is therefore available to the community if necessary. We have added a paragraph on H3K9me2 and H3K9me3 to the discussion in response to reviewer 2 (see below) in which we mention the ChIP-seq experiment with these PTMs and provide the accession number for the data.

We did not feel that the omission of data for H3K9me2 and H3K9me3 was problematic because the manuscript focused principally on the association of histone PTMs with genes and there was already a lot of data to present and a lot of interesting observations to discuss for that part of the analysis. Clearly, however, both of the reviewers are interested in having more information about the non-coding part of the genome and we have now added additional information about this aspect in relation specifically to H4K20me3 (see below).

3. Although there is the note "Almost all (94.6%) of the H4K20me3 peaks were associated with repeated sequences or transposons, predominantly in intergenic regions but also in introns." with a much longer commentary on that note in the Discussion, there are no results shown (except a vague mention of Fig.3). The report lacks investigation of the non coding part of the genome including repeats and transposable elements that have been annotated by the Authors in previous work. This should be added here, because the Authors must have the data and the report is incomplete without this. Especially because the abstract claims "The overview of the nature and functions of histone PTMs in the brown algae presented here will provide a foundation for future studies"

Reply: We have extended the analysis of the non-coding part of the genome in the revised manuscript, concentrating on H4K20me3, which we had already shown to be strongly associated with transposons and repeated sequences in the non-coding part of the genome. We have added the following information for H4K20me3:

- For figure 2a, which shows the distribution of each histone PTM across different features of the genome, the proportions for intergenic regions, introns and first introns have now been broken down into features that include at least one inserted transposon and features without inserted transposons. This visualisation highlights the strong association of H4K20me3 with non-coding regions containing transposons and we have also confirmed the strong preferential association with transposon-containing introns and intergenic regions using a statistical test, which is reported in the modified manuscript.
- A new panel in figure 2 (figure 2d) shows distributions all the histone PTM along a representative region of the genome on which we have indicated both genes and transposons. Again this panel illustrates the association of H4K20me3 with transposons.
- An analysis of H4K20me3 peaks across different TE families indicated that it essentially marks all classes of transposon.
- Given that H4K20me3 was found to be associated with transposons in both intergenic regions and in non-coding regions of genes (introns, figure 2a), we also looked at the relationship between the presence of H4K20me3 and gene feature parameters such as exon number, gene length, gene expression levels and the presence of TSS-localised marks (new figure 4). Interestingly, genes that harboured a H4K20me3 peak exhibited significantly weaker signals for all five of the TSS-localised PTMs (H3K4me2, H3K4me3, H3K9ac, H3K14ac and H3K27ac). This observation is discussed in the revised version of the manuscript.
- We have identified a strong correlation between the presence of H4K20me3 and the positions of H3K79me2 domains throughout the genome (figure 5a,b, previously figure 4). This observation is interesting because it support the hypothesis, proposed in the discussion, that the H3K79me2 marked regions may extend to neighbouring TSSs in order to regulate transcriptional activity in genomic regions containing transposons (lines 452-459).
- Interestingly, analysis of loci marked by H4K20me3 but not H3K79me2 or with H3K79me2 but not H4K20me3 showed that both PTMs are associated with reduced gene expression and reduced presence of the TSS-localised PTMs H3K4me2, H3K4me3, H3K9ac, H3K14ac and H3K27ac (new figure S6). Moreover, the two PTMs have an additive, negative effect on both gene expression and the deposition of the TSS-localised PTMs (new figure S6), i.e. both parameters are even more strongly reduced when both PTMs are present.

4. the title "epigenetic reprogramming during ectocarpus life cycle" is misleading. First none of the marks studied here are "epigenetic" marks (only K27me3 and K9me2/3 are defined as such when the term is used properly). Second what is shown here reflects the change of transcription program, but has nothing in common with epigenetic reprogramming such as described in mammals where an histone modifications or DNA methylation is erased and reinstalled. Here as quoted by the Authors "Overall, the distribution of PTMs was stable between the two life cycle generations." And "there were not any major, genome-wide changes in patterns of histone PTMs associated with alternation between life cycle generations.:" These two statements contradict the very notion of reprogramming and this term should not be used, because it implies very broad changes of patterns of marks triggered by events other than transcription. Please remove the term reprogramming and moderate the statement both in abstract, introduction, result and discussion.

Reply: We agree with the reviewer's remark. The term "epigenetic reprogramming" was used incorrectly. We have rephrased the manuscript throughout. The relevant results and discussion section now focus on the stability of histone PTM patterns between life cycle generations (sporophyte and gametophyte), which we nonetheless consider to be an interesting and novel observation. There are still some sentences that use the term "epigenetic processes", where it is meant in the sense of chromatin-related processes that do not involve modification of the DNA sequence. If the reviewer considers that the term is used incorrectly could they please propose an alternative?

Reviewer #2:

The manuscript by Bourdareau et al. reports on a series of genomics, proteomics and bioinformatics analyses of histone modifications in the brown alga *Ectocarpus*. They provide novel and convincing data on the presence and distribution of a multitude of likely histone modifications, including several new histone modifications determined through proteomics analyses of histone preparations from *Ectocarpus*. It is a shame that no antibodies exist against these 'new' histone modifications, such as H2A.ZK9ac or H2A.ZK20ac/me so that their relative amounts could be ascertained and their genome wide distribution be determined.

The authors did apply extensive ChIP-seq analyses to generate genome-wide maps of eight histone marks often assayed in many other eukaryotes. Five of these marks, H3K4me2, H3K4me3, H3K9ac, H3K14ac and H3K27ac were shown to be distributed within transcriptionally active regions of the *Ectocarpus* genome, and significantly depleted from the gene bodies while being enriched around the TSS. These observations, together with the observed genome-wide distribution of H3K36me3 mark, are in agreement with a rich body of literature across all eukaryotic lineages showing their association with transcriptionally active regions, and bring little novelty to chromatin modification distribution field, but may be of interest to comparative epigenomics field.

Reply: We understand the point that reviewer is making but we feel that it is important to emphasise that this study concerns a lineage that has been evolving independently of plants and animals for over a billion years and, also, has independently evolved complex multicellularity. Therefore, the identification of chromatin features that are conserved with plants and animals is not trivial from an evolutionary perspective in as far as it supports extremely deep evolutionary origins for these features, stretching back to the crown radiation of the eukaryotes. Of course, comparisons between plants and animals have already provided

evidence for the ancient origins of many of these features but the inclusion of additional lineages such as the brown algae is considerably enriching our understanding of this evolutionary history. We believe therefore that the observations presented in the manuscript are of considerable interest in terms of comparative epigenomics and that the features that are conserved with plants and animals are probably as interesting as those that differ (such as the loss of polycomb complexes for example).

On the other hand, H4K20me3 mark was found associated with silent regions, mostly transposon-rich domains, but, interestingly in gene introns as well. The authors explain this by the fact that intron sequences comprise 40% of the *Ectocarpus* genome. I presume the authors' assumption there is that repeats and remnants of repeats are also likely to be found in many intron sequences? It'd be worth saying that outright, especially if it can be backed up by information from the *Ectocarpus* genome analysis that the authors should be quite familiar with.

Reply: We now state in the text that "44.0% of the transposons in the genome are located in introns." The revised version of the manuscript includes a more detailed analysis of H4K20me3 in response to reviewer 1, including the association of this PTM with transposons both in intergenic regions and introns. We have also examined in more detail the effect of genic (intronic) H4K20me3 peaks on gene expression and other gene parameters.

One recommendation I could make here would be to include a discussion on what roles might H3K9 methylation have in the silencing of these regions, in the absence of empirical data on H3K9me2/3 presence and distribution in *Ectocarpus*.

Reply: We are reluctant to speculate about the possible roles of H3K9me2 and H3K9me3 in *Ectocarpus* in the absence of profile data for these PTMs but we have added the following paragraph to the discussion section to try to reply to the reviewer's request: "Heterochromatin is marked by methylation of lysine 9 of histone H3 in animals and land plants [55]. H3K9me3 is the most abundant heterochromatin marker in animals [56,57]. In plants, heterochromatin is silenced by a feedback loop involving H3K9me2 and DNA methylation [58], whereas H3K9me3 is associated with euchromatin and appears to have a different function [46]. H3K9me2 and H3K9me3 were detected in *Ectocarpus* but appear to be present at low abundance and ChIP-seq profiling did not provide any exploitable information about the distribution of these PTMs (unpublished results, data available at NCBI Gene Expression Omnibus accession GSE146369). More sensitive analysis methods will be necessary to investigate the roles of H3K9me2 and H3K9me3 in brown algae but it is possible that other PTMs, such as H4K20me3 and H3K79me3, carry out at least some of the functions that have been attributed to these PTMs in other lineages."

Interestingly, the authors show that H3K79me2 mark can be found in large genomic blocks of several kb, often spanning several genes, altogether spanning a combined 1/3 of the genome. Intriguingly, while the genes contained within the H3K79me2 blocks are less transcriptionally active and have an overall lower active gene marks (like H3K4me3 /H3K9ac / H3K27ac) the borders of these H3K79me2 domains seem to be defined by TSS of genes flanking the domain, implying that some sort of transcriptional interference with the establishment or

spreading of the H3K79me2 mark. The authors make a nice comparison with the contrasting distribution of this mark in animals, and some of the ideas they presented in the discussion are nice food for thought. It is unfortunate that there is no genetics in *Ectocarpus* that would enable testing some of their models and hypotheses.

Reply: Note that we have added an additional supplementary figure (figure S5) with a statistical test to further support the co-localisation of H3K79me2 domain borders and TSSs/TESSs.

The authors also make a convincing argument using proteomic and genomics data that both PRC2 and PRC1 complexes as well as H3K27me3 mark are absent from *Ectocarpus*, and therefore conclude that Polycomb silencing is missing from *Ectocarpus* repertoire of gene silencing mechanisms.

The section on 'epigenetic reprogramming' during the *Ectocarpus* life cycle is probably the weakest component of the manuscript, since the authors provide little evidence of any truly epigenetic reprogramming between the sporophyte and gametophyte. While they do identify 774 genes that exhibited generation-biased pattern of gene expression (including, interestingly, a significant enrichment of lncRNA in this set, something that could be analyzed in greater depth during revision, especially having in mind huge correlation in coordinated regulation of lncRNA and its adjacent coding gene), only a third of these genes show a change in chromatin state (as defined by a set of histone H3 modifications). While the causal relationship between gene expression and the chromatin marks analyzed is interpreted as 'histone-first' model ("epigenetics"), it is worth pointing out that the act of transcription itself may be what is recruiting these histone modifications (rather than the other way around), and therefore making the epigenetic argument very weak. I would recommend this section be rewritten and perhaps re-focused.

Reply: We agree with the reviewer that the use of the term epigenetic reprogramming in this context was incorrect. The sections on 'epigenetic reprogramming' during the *Ectocarpus* life have been modified in both the results section and the discussion and all references to epigenetic reprogramming have been removed, including from the abstract.

In this context, however, we also feel that it is important to stress that the similarity between histone PTM patterns during the sporophyte and the gametophyte generations is not a trivial observation. Comparative analyses of life cycle generations (sporophyte and gametophyte) have not been carried out in other systems and we believe that the high level of stability of histone PTM patterns between generations is an interesting and important observation.

We have also modified the discussion of the relationship between histone PTMs and gene expression to include both of the possibilities discussed by the reviewer: that the histone PTMs influence transcription or that the transcription may lead to modifications of the histone PTM pattern. We have changed the text throughout the manuscript to take into account this modification, in particular we present the observed patterns as correlations between PTMs and gene expression rather than directly inferring any functional role. However, we would like to point out that, even if it is transcription that leads to the recruitment of these histone marks, it is likely that they play a role in the regulation of gene expression at some level (other than the initial induction of gene expression), for example by stabilising the gene in a transcriptionally active state. We have preferred not to speculate along these lines in the manuscript but include this comment here to underline that both

scenarios (PTM activation of transcription or transcriptional activation of PTM deposition) are potentially of interest.

The reviewer is correct in stating that only a third of the GBGs showed a change in chromatin state based on the histone PTMs analysed but we noted additional correlations between gene expression and chromatin state such as the tendency for larger TPM fold changes to be associated with transitions to chromatin state 1 and the tendency of the GBGs with the highest fold changes to also exhibit changes in chromatin state. These observations provide additional support for a causal relationship between chromatin state and gene expression (although not the direction of the causal relationship). We therefore believe that the fact that chromatin state transitions were detected for only a third of the GBGs should be interpreted more as an indication of the complexity of the system than as a lack of a causal relationship.

Finally, as far as the lncRNAs are concerned, it was not clear to us what additional analysis the reviewer would like us to carry out. We have carried out an analysis of the distribution of histone PTMs at lncRNA loci (figure 2c, lower panel) and an extensive analysis of relationships between differentially expressed lncRNAs and adjacent, differentially expressed protein-coding genes (supplementary table S5). Unfortunately, we have very little information about the functions of lncRNAs in *Ectocarpus* and it is therefore difficult to see how we could extend this analysis. Of course, we would be very happy to do so if the reviewer has some specific suggestions in mind.

Overall, this manuscript reports on a series of ChIP-seq, RNA-seq and proteomics data that are descriptive in nature, and while well the experiments were well executed and contribute a large new body of data to the literature, it does not allow for any causal/functional inferences to be made. This is always an issue with working on orphan model organisms with limited genetic toolkit, and I would encourage the authors to revise their manuscript as indicated throughout this review in order to make it suitable for publication in *Genome Biology*.

As a result of the significant disruption that is being caused by the COVID-19 pandemic we are very aware that many researchers will have difficulty in meeting the timelines associated with our peer review process during normal times. Please do let us know if you need additional time. Our systems will continue to remind you of the original timelines but we intend to be highly flexible at this time.

Second round of review

Reviewer 1

The Manuscript improved a lot and the Authors have clarified certain points but some answers are still incomplete or raise additional questions. Basically, findings are largely confirmatory and the presence or absence of PRC2 and H3K27me3 in brown algae remains unclear because the mark is apparently detected in Western blot but not in Mass spectrometry and the Authors show no sign of PRC2 in *Ectocarpus* but do not investigate genomes of other brown algae to reach a conclusion on the presence or absence of PRC2 in brown algae. The major novelty of the report is the PTM H3K79me2 and the Authors should pursue a more detailed analyses of gene encoding Set domain proteins to highlight what could be the gene responsible for this mark. There are still a few additional points outlined to clarify before the paper is published.

Major comments:

1. Fig1b shows that nearly all marks specific to *Ectocarpus* to the exception methylation of H4 and H3K79me2, are on H2A.Z. Concerning these marks, their “specificity” likely arises from the divergence of H2A.Z itself. The Authors should investigate this point based on H2A.Z protein sequences alignments. If indeed changes in H2A.Z sequence is the source of the marks, the Authors should not highlight these as new marks but the sole consequence of H2A.Z divergence.
2. It is interesting to note that the H3K36me3 profile over genes in *Ectocarpus* resembles more the profiles reported in animals than in plants. The Authors should extend this type of comparison to other marks and to fungi using *Neurospora* as a model. This should be included in the analysis presented in Fig1b
3. The abundance of H3K9 methylation is still unclear. The Western blots shown in FigS3 lack a control using a species where the marks are abundant, animals or plant or yeast could be used easily. In addition H3K9me1 is a major marker of plant heterochromatin and should be investigated as was done in FigS9. Western blots for H3K9me1, me2 and me3 should be provided with controls using animal or plant cells.
4. The absence of PRC1 and PRC2 encoding genes in *Ectocarpus* is clear but it is unclear why H3K27me1/2/3 are reported in Fig1? The Authors should also report the presence or absence of these genes in the other genomes of brown algae that are currently available, *Desmarestia herbacea* (doi: 10.1186/s13059-020-02041-z from the Authors own work), *Cladosiphon* (DNA Res. 2016 Dec;23(6):561-570. doi: 10.1093/dnares/dsw039) and *Saccharina japonica* (Nature Communications volume 6, Article number: 6986 (2015)). Alternatively if the Authors do not wish to perform this search they should be careful to outline that the absence of PRC2 might also be specific to *Ectocarpus* in their conclusion. For example it was for a long time misleading to think that PRC2 is not present in fungi due to conclusion based on model yeast Yet it was shown later that PRC2 is present in other ascomycetes and in basidiomycetes.
5. Table S2 should include *Arabidopsis* and *Neurospora* for additional comparison and should highlight what SET domains proteins are truly unique to *Ectocarpus* and these might indeed be ascribed as potential writers of H3K79me2.

Minor comments:

1. Line 284 “In most cases (84%), the lncRNA and its adjacent protein-coding gene were co-ordinately upregulated during the same generation of the life cycle. These observations suggest a possible role for lncRNAs in the regulation of adjacent genes in *Ectocarpus*.” It is equally possible that it happens that adjacent genes are under the same type of control. For a lncRNA to act there should be a matching target sequence on the gene regulated and the Authors should look for such a target to support their hypothesis.
2. Line 301 “that it is also possible that some or all of the observed histone PTM modifications were a consequence, rather than a cause, of transcription.” Indeed, most PTM are the consequence of transcriptional activity and the few exceptions are the “repressive” PTMs, which are apparently missing in *Ectocarpus* H3K9me2/3 and H3K27me3. However, the Authors should consider H3K79me2 that marks blocks of chromatin as a potential replacement of the “repressive” PTMs.
3. Line 486 Please remove or edit this statement with a strikethrough editing mark “This stability was consistent with the observation that only about 4% of genes, genome-wide, were significantly differentially expressed between the two generations.”

Reviewer 2

The revised version of the manuscript by Bourdareau is improved and it is now suitable for publication following minor corrections. The wealth of data and analyses in it will certainly be welcome by the researchers in the field of comparative epigenomics.

I understand the authors' comments regarding gene and histone PTM changes between generations, but I still think that this section is the weakest one in the manuscript and may be enhanced by clarifying the functional categories of genes in the list of ~700 GBG. (any GO term enrichment for functions?).

New figure legends need some careful editing (for example S3 legend needs clarification/correction on the labels used in the text vs the actual blot images).

Reviewer reports:

Reviewer #1: The Manuscript improved a lot and the Authors have clarified certain points, which now raises additional questions. Basically, findings are largely confirmatory and the presence or absence of PRC2 and H3K27me3 in brown algae remains unclear because the mark is apparently detected in Western blot but not in Mass spectrometry and the Authors show no sign of PRC2 in *Ectocarpus* but do not investigate genomes of other brown algae to reach a conclusion on the presence or absence of PRC2 in brown algae. The major novelty of the report is the PTM H3K79me2 and the Authors should pursue a more detailed analyses of gene encoding Set domain proteins to highlight what could be the gene responsible for this mark. There are still a few additional points outlined to clarify before the paper is published.

Reply: We agree with the reviewer that there is some ambiguity regarding the presence of H3K27me3 because this PTM was not detected by mass spectrometry but we did detect a very weak signal when an immunoblot was carried out with an anti-H3K27me3 antibody (see reply below concerning the immunoblots). However, given that PRC2 is clearly absent from *Ectocarpus* and that ChIP-seq experiments using the anti-H3K27me3 antibody failed to detect any specific chromatin immunoprecipitation products (GEO accession number GSE146369), we believe that the most probable interpretation of these results is that the anti-H3K27me3 antibody bound weakly and non-specifically to a non-H3K27me3 antigen in the *Ectocarpus* extracts. We have therefore concluded that H3K27me3 is absent. We understand that these considerations are quite complicated but we feel that it is important that all the observations concerning this mark are presented honestly, even if this makes the story a bit more difficult to follow for the reader.

See below for our replies to the reviewer's comments concerning the analysis of the presence of PRC genes in other brown algal genomes and the link between SET domain proteins and H3K79me2.

Major comments:

1. Fig1b shows that nearly all marks specific to *Ectocarpus* to the exception methylation of H4 and H3K79me2, are on H2A.Z. Concerning these marks, their "specificity" likely arises from the divergence of H2A.Z itself. The Authors should investigate this point based on H2A.Z protein sequences alignments. If indeed changes in H2A.Z sequence is the source of the marks, the Authors should not highlight these as new marks but the sole consequence of H2A.Z divergence.

Reply: This point was not raised during the initial evaluation but we are happy to have had the chance to carry out a more extensive analysis of H2A.Z, which has allowed us to correct an error in the previous versions of the manuscript. But first we would like to remind the reviewer that H3K79me2 has been detected in animals [1] and, therefore, was not included in the list of *Ectocarpus*-specific PTMs shown in Fig. 1b. As far as the PTMs on H2A.Z are concerned, Fig. 1b listed six *Ectocarpus*-specific PTMs, of which only two (H2A.ZK20ac and H2A.ZR38me1) were associated with H2A.Z. We originally indicated in Table S3 that there was no equivalent residue for either of these PTMs in the human H2A.Z proteins but we have now carried out a more comprehensive alignment of H2A.Z proteins and this new analysis indicates that residues K20 and R38 of the *Ectocarpus* H2A.Z protein are in fact homologous to residues K13 and R31, respectively, of the two human H2A.Z proteins (see Reviewer Fig. 1 below). Based on this analysis, the *Ectocarpus* PTM H2A.ZK20ac appears to be conserved in humans (human H2A.ZK13ac) and we have removed this PTM from the list of *Ectocarpus*-specific PTMs in the bottom part of Fig. 1b. Table S3 has also been corrected. In conclusion, therefore, only one H2A.Z PTM is unique to *Ectocarpus*. Note that this uniqueness is not a consequence of H2A.Z sequence divergence because the residue is conserved in diverse H2A.Z proteins.

Reviewer Fig. 1. Alignment of the amino-terminal ends of diverse histone H2A.Z proteins showing conservation of Ec-26_6500 residues K20 and R38.

2. It is interesting to note that the H3K36me3 profile over genes in *Ectocarpus* resembles more the profiles reported in animals than in plants. The Authors should extend this type of comparison to other marks and to fungi using *Neurospora* as a model. This should be included in the analysis presented in Fig1b

Reply: In the discussion section we have discussed the results obtained for all eight of the PTMs assayed in *Ectocarpus* in the context of equivalent studies carried out on model systems from both the animal and land plant lineages. In each case, the pattern observed in *Ectocarpus* is compared with the current knowledge about each individual PTM for the other two lineages. We chose to focus on the animal and land plant lineages because both are well characterised and because, like the brown algae, both include multicellular organisms with complex developmental programs (animals, land plants and brown algae represent the three most complex of the multicellular lineages). We feel that it would unnecessarily complicate the text if we were to include multiple comparisons with additional model systems but fungi are obviously relevant in this context and we have now added references that cite work on fungal systems. We would be happy to cite additional features of fungal systems if the reviewer thinks that these would be particularly relevant to this study or if they might help explain specific patterns we have observed.

3. The abundance of H3K9 methylation is still unclear. The Western blots shown in FigS3 lack a control using a species where the marks are abundant, animals or plant or yeast could be used easily. In addition H3K9me1 is a major marker of plant heterochromatin and should be investigated as was done in FigS9. Western blots for H3K9me1, me 2 and me3 should be provided with controls using animal or plant cells.

Reply: Figure S3 has been updated and now includes immunoblots for H3K9me1, H3K9me2 and H3K9me3, all with HeLa cell extract positive controls. The HeLa controls confirm that the signals for H3K9me2 and H3K9me3 are weak in *Ectocarpus*.

We also detected a weak signal for H3K9me1 but this mark was not detected by mass spectrometry. It is therefore possible that the signal is non-specific. In any event, the weak signal does not support a major role for H3K9me1 as a heterochromatin marker in *Ectocarpus*.

The updated figure also includes immunoblots for H3K27me1, H3K27me2 and H3K27me3, along with HeLa cell extract positive controls. None of these PTMs were detected in *Ectocarpus* chromatin extracts. As mentioned in the text (and in the updated figure legend), a weak signal can be detected for H3K27me3 after long exposure times but, overall, the evidence (absence of

a signal in the mass spectrometry data and in the ChIP-seq experiments and absence of PRC2 complex genes in the genome) indicates that this PTM is absent and therefore that the immunoblot signal is probably non-specific.

Fig. 1a and Table S3 have been updated based on the results of these experiments.

4. The absence of PRC1 and PRC2 encoding genes in *Ectocarpus* is clear but it is unclear why H3K27me1/2/3 are reported in Fig1? The Authors should also report the presence or absence of these genes in the other genomes of brown algae that are currently available, *Desmarestia herbacea* (doi: 10.1186/s13059-020-02041-z from the Authors own work), *Cladosiphon* (DNA Res . 2016 Dec;23(6):561-570. doi: 10.1093/dnares/dsw039) and *Saccharina japonica* (Nature Communications volume 6, Article number: 6986 (2015)). Alternatively if the Authors do not wish to perform this search they should be careful to outline that the absence of PRC2 might also be specific to *Ectocarpus* in their conclusion. For example it was for a long time misleading to think that PRC2 is not present in fungi due to conclusion based on model yeast Yet it was shown later that PRC2 is present in other ascomycetes and in basidiomycetes.

Reply: In the previous, revised version of the manuscript we only reported H3K27me2 in Fig.1 (and Table S3). Is it possible that the reviewer confused K27 with another residue in this figure? H3K27me3 was removed from both the figure and the table during the first revision because, taken together, our analyses indicate that this PTM is absent in *Ectocarpus*. We stated this in the legend to Table S3 and in the text.

H3K27me2 was detected by mass spectrometry (Fig. S2) and therefore appears to be present in *Ectocarpus* despite the absence of PRC2. Presumably it is deposited by another, as yet unidentified, enzyme. Given the very large phylogenetic distances relative to animals and land plants, it is possible that the same PTM is deposited by different enzymes in different lineages. There is a precedent for this for H3K27me1, for example, which is deposited by E(Z) in mammals but by ATXR5 and ATXR6 in plants [2]. In general, therefore, we feel that the data presented in Fig. 1 and Table S3 should only take into account evidence directly pertaining to the presence or absence of individual PTMs, independent of the presence or absence of genes encoding putative PTM writers in the *Ectocarpus* genome.

We have analysed the three additional published brown algal genomes (*Cladosiphon okamuranus*, *Nemacystus decipiens* and *Saccharina japonica*) for the presence of PRC genes (revised Table S7, now Table S10). We did not analyse the *Desmarestia herbacea* genome because, although the Baudry *et al.* paper reported a high quality structural assembly, there is not yet a good quality gene annotation for this genome. Analysis of the three new genomes, which cover two major brown algal orders separated by at least 100 My of evolution, indicated that they all share exactly the same set of PRC and PRC-associated genes as was observed for *Ectocarpus*. In particular, the analysis indicated that all four species lack all the core components of both the PRC2 and the PRC1 complexes (with the exception of RbAp48). This analysis therefore supported our proposal that loss of PRC2 and PRC1 predated the divergence of the brown algae from other stramenopile lineages. A sentence stating that a total of four brown algal genomes were found to lack genes encoding the PRC core proteins has been added to the Results section.

5. Table S2 should include *Arabidopsis* and *Neurospora* for additional comparison and should highlight what SET domains proteins are truly unique to *Ectocarpus* and these might indeed be ascribed as potential writers of H3K79me2.

Reply: This is a new request, that was not made during the initial evaluation. We agree that it would be interesting to add data for a land plant and a fungus to this table but that would require a considerable investment in terms of analysis of the literature and we believe that this sort of analysis would be more appropriate for a review article. As it stands, the table already allows the reader to see which *Ectocarpus* proteins can be equated with a human (or a diatom) protein and which ones are either novel or difficult to assign to a specific enzymatic group. This was our intention with the table and we do not believe that adding the additional species would significantly improve it as far as this

function is concerned. If, however, the reviewer has strong arguments for including the additional species based on published information about specific chromatin modifier proteins, we would be willing to invest the time required to add the additional data.

As far as the second part of the comment is concerned, it is not clear to us why the writer of H3K79me2 in *Ectocarpus* should be a protein that is found uniquely in this lineage? Given that animals also have H3K79me2 (see reply to comment 1), it seems more likely that one of the four DOT1-domain-containing KMTs encoded by the *Ectocarpus* genome (Table S2) is responsible for H3K79 dimethylation (although, of course, we cannot formally rule out the possible hypothesis that different enzymes deposit H3K79me2 in animals and brown algae - see reply above). In any event, Table S2 already provides information about which of the *Ectocarpus* SET domain proteins are predicted to be orthologous to proteins in the two other species.

Minor comments:

1. Line 284 "In most cases (84%), the lncRNA and its adjacent protein-coding gene were coordinately upregulated during the same generation of the life cycle. These observations suggest a possible role for lncRNAs in the regulation of adjacent genes in *Ectocarpus*." It is equally possible that it happens that adjacent genes are under the same type of control. For a lncRNA to act there should be a matching target sequence on the gene regulated and the Authors should look for such a target to support their hypothesis.

Reply: We agree that there may be several explanations for the juxtaposition of lncRNA and mRNA GBGs. We just wanted to suggest this possibility as it would be particularly interesting and worth following up on in the future. The phrasing we used (with "suggest" and "possible") was meant to convey that we did not propose this hypothesis as the sole possible explanation. We also agree that one might expect a sequence match between the lncRNA and the mRNA transcripts if the former regulates the latter, although it is also possible to imagine mechanisms that do not involve direct interactions between nucleic acids. Moreover, if there is a match, it may be difficult to detect bioinformatically because it may not involve a region of complete identity and interaction could be influenced by factors such as secondary structure, free energy of binding and protein cofactors. Consequently, lncRNA target genes are usually identified experimentally, either by analysing transcript abundances after lncRNA knockdown or using specific binding assays (pull-down assays, luciferase reporter assays or immunoprecipitation assays), approaches that are beyond the scope of this study. Within the context of this paper, we have attempted to address the reviewer's comment by adding a supplementary table (new Table S9) that provides more information about the adjacent lncRNA GBG and mRNA GBG pairs and in which we have reported the results of an EMBOSS Matcher analysis (length and score of match) for each lncRNA-mRNA pair. This analysis identified very strong matches for some lncRNA GBG and mRNA GBG pairs, including, of course, those for which the transcribed regions actually overlap. For some other pairs, matches were weaker but we preferred to report all matches along with scores rather than impose an arbitrary cut-off.

2. Line 301 "that it is also possible that some or all of the observed histone PTM modifications were a consequence, rather than a cause, of transcription." Indeed, most PTM are the consequence of transcriptional activity and the few exceptions are the "repressive" PTMs, which are apparently missing in *Ectocarpus* H3K9me2/3 and H3K27me3. However, the Authors should consider H3K79me2 that marks blocks of chromatin as a potential replacement of the "repressive" PTMs.

Reply: Based on the remarks from reviewer 2, we were very careful in the previous, revised version of the manuscript to systematically consider the possibility that histone PTM modifications may be deposited as a consequence of transcription, in addition to discussing the converse hypothesis.

The possibility that H3K79me2 that marks blocks of chromatin as a potential replacement for PTMs that are "repressive" in other lineages was evoked in the paragraph that we added to the

discussion section in response to a comment from reviewer 2 during the first round of revision: "More sensitive analysis methods will be necessary to investigate the roles of H3K9me2 and H3K9me3 in brown algae but it is possible that other PTMs, such as H4K20me3 and H3K79me3, carry out at least some of the functions that have been attributed to these PTMs in other lineages."

3. Line 486 Please remove or edit this statement with a strikethrough editing mark "This stability was consistent with the observation that only about 4% of genes, genome-wide, were significantly differentially expressed between the two generations."

Reply: We had intended to remove this sentence from the revised manuscript. It has now been completely deleted.

Reviewer #2: The revised version of the manuscript by Bourdareau is improved and it is now suitable for publication following minor corrections. The wealth of data and analyses in it will certainly be welcome by the researchers in the field of comparative epigenomics.

I understand the authors' comments regarding gene and histone PTM changes between generations, but I still think that this section is the weakest one in the manuscript and may be enhanced by clarifying the functional categories of genes in the list of ~700 GBG. (any GO term enrichment for functions?).

Reply: We have added an analysis of the predicted functions of the GBGs to the revised manuscript. The analysis used a system of manually-assigned functional categories [3] rather than GO terms because the latter were established principally with animal protein function in mind and are often poorly suited for brown algal protein sets. We also predicted the subcellular locations of the GBG proteins using the Hectar algorithm [4]. The complete results of the analyses have been added to Table S5 (see column H) and statistical tests are presented in new Tables S6 and S7.

Furthermore, in light of the points raised by the reviewers (and agree by the authors in their rebuttal), it may be prudent to re-word the first subtitle of the Discussion from "Epigenetic regulation in..." to something less glaringly unsupported by the results (ie there is no evidence of any truly epigenetic regulation in their results -- it is all about chromatin regulation and gene expression as correlates).

Reply: We have modified the subtitle to "Histone post-translational modifications in a multicellular brown alga".

New figure legends need some careful editing (for example S3 legend needs clarification/correction on the labels used in the text vs the actual blot images).

Reply: The figure legend has been corrected. In addition, we have added the following sentence to figures S3 and S9: "Bars to the left of gel images indicate the positions of molecular size markers. kDa, kilodalton."

Additional improvements: We have also added an additional supplementary table to the revised manuscript that summarises histone PTMs detected at all the genes in the *Ectocarpus* genome, together with transcript abundances based on RNA-seq data (Table S13).

Reviewer's Responses to Questions

Evaluation. Has the author satisfactorily responded to your previous review?

Reviewer #1: No

Reviewer #2: Yes

References

1. Steger DJ, Lefterova MI, Ying L, Stonestrom AJ, Schupp M, Zhuo D, et al. DOT1L/KMT4 recruitment and H3K79 methylation are ubiquitously coupled with gene transcription in mammalian cells. *Mol Cell Biol.* 2008;28:2825–39.
2. Jacob Y, Michaels SD. H3K27me1 is E(z) in animals, but not in plants. *Epigenetics.* 2009;4:366–9.
3. Arun A, Coelho SM, Peters AF, Bourdareau S, Pérès L, Scornet D, et al. Convergent recruitment of TALE homeodomain life cycle regulators to direct sporophyte development in land plants and brown algae. *Elife.* 2019;8:e43101.
4. Gschloessl B, Guerneur Y, Cock J.M. HECTAR: a method to predict subcellular targeting in heterokonts. *BMC Bioinf.* 2008;9:393.